# Pathogenic shifts in endogenous microbiota impede tissue regeneration via distinct activation of TAK1/MKK/p38

**Christopher P Arnold[1], M Shane Merryman[1], Aleishia Harris-Arnold[1], Sean A McKinney[1], Chris W Seidel[1], Sydney Loethen[2], Kylie N Proctor[3], Longhua Guo[1], Alejandro Sánchez Alvarado[4]***

[1]Stowers Institute for Medical Research, Kansas City, United States; [2]University of Missouri, Columbia, United States; [3]Pittsburg State University, Pittsburg, United States; [4]Howard Hughes Medical Institute, Stowers Institute for Medical Research, Kansas City, United States

**Abstract** The interrelationship between endogenous microbiota, the immune system, and tissue regeneration is an area of intense research due to its potential therapeutic applications. We investigated this relationship in *Schmidtea mediterranea*, a model organism capable of regenerating any and all of its adult tissues. Microbiome characterization revealed a high Bacteroidetes to Proteobacteria ratio in healthy animals. Perturbations eliciting an expansion of Proteobacteria coincided with ectopic lesions and tissue degeneration. The culture of these bacteria yielded a strain of Pseudomonas capable of inducing progressive tissue degeneration. RNAi screening uncovered a TAK1 innate immune signaling module underlying compromised tissue homeostasis and regeneration during infection. TAK1/MKK/p38 signaling mediated opposing regulation of apoptosis during infection versus normal tissue regeneration. Given the complex role of inflammation in either hindering or supporting reparative wound healing and regeneration, this invertebrate model provides a basis for dissecting the duality of evolutionarily conserved inflammatory signaling in complex, multi-organ adult tissue regeneration.

*For correspondence: asa@stowers.org

## Introduction

Host and resident microorganism interactions are not only integral to the maintenance of normal physiological functions, but also to the development of pathology. Microbial dysbiosis, for example, underlies persistent inflammatory disorders, chronic non-healing wounds, and scar formation (*Dowd et al., 2008*; *Price et al., 2009*; *Carvalho et al., 2012*; *Scales and Huffnagle, 2013*; *Brothers et al., 2015*; *Shin et al., 2015*). Long-term management of these conditions constitutes a substantial and sharply rising burden on our healthcare system (*Sen et al., 2009*). Significant evidence spanning a diverse array of organisms and tissues indicates that the immune response can play a central role in either promoting or hindering wound healing and tissue regeneration (*Eming et al., 2009*; *Karin and Clevers, 2016*). The precise determinants facilitating these diametrically opposed outcomes is an area of intense investigation. As such the development of diverse and robust model systems capable of elucidating basic molecular mechanisms governing the interplay of the immune response and tissue regeneration should provide new insights for the design of novel therapies to combat this rising medical issue (*Sen et al., 2009*).

Planaria are a classic model system for studying adult wound healing and tissue regeneration (*Reddien et al., 2004*). These free living members of the phylum Platyhelminthes contain a persistent pool of adult pluripotent stem cells, termed neoblasts, capable of regenerating all of the tissues

**eLife digest** Regeneration, the ability to replace missing or damaged tissue, has fascinated biologists for years and has inspired a new direction for the medical field. Figuring out how some animals easily accomplish this while others do not may help us to develop new therapies that enhance regeneration in humans.

Previous work has indicated that the immune system, which is normally used to defend the body against bacteria, plays an important but complicated role in regeneration. By studying the relationships between bacteria, the immune system and regeneration in simple systems, it may be possible to see how their interactions either support or prevent the replacement of lost tissues.

Flatworms called planaria can regenerate all of their tissues. Arnold et al. have now investigated what bacteria exist in planaria, how the planarian immune system responds to these bacteria, and how this response affects regeneration. The results reveal that the two main types of bacteria that are present in planaria are also found in humans. In fact, conditions that encourage the growth and spread of one of these types of bacteria (called Proteobacteria, many of which can make humans ill) damaged the worms and prevented them from regenerating.

Arnold et al. then looked to see if the worms had genes that were similar to human genes that control the key immune process of inflammation, and found evidence of several such genes. Reducing the activity levels of these genes enabled worms that had been infected with Proteobacteria to regenerate again. However, these genes only seem to be responsible for regeneration when the planaria are infected with bacteria. Thus, planaria could be used as a simple model to discover how changes in resident bacteria can be detected by the immune system and affect the ability to regenerate tissues.

Future studies could use planaria to identify even more genes that control regeneration during infection. Also, since the main types of bacteria in planaria are similar to those in humans, planaria may help us to learn how animals can properly balance the levels of these bacteria in order to remain healthy.

and cell types of the organism (*Wagner et al., 2011*). Ablation of this population of mitotic cells via irradiation eliminates regenerative capabilities, resulting in regression of anterior structures and eventual tissue lysis (*Reddien et al., 2005*). Development of RNAi methodologies has enabled the interrogation of genes involved regeneration (*Sanchez Alvarado and Newmark, 1999*; *Reddien et al., 2005*). These studies have informed our understanding of how wound responses, the recognition of lost tissues, dynamic establishment of positional identity, and activation of appropriate stem cell and differentiation programs all serve to accomplish large-scale complex tissue regeneration (*Wenemoser and Reddien, 2010*; *Petersen and Reddien, 2011*; *Wenemoser et al., 2012*; *Scimone et al., 2014*). While wounding is the stimulus for regeneration across all organisms studied to date, its incidence also presents an additional challenge. Disruption of barrier epithelia and exposure of mucosal surfaces poses an increased risk for bacterial invasion of internal tissues. Yet the role to which, if any, changes in endogenous microbiota or the planarian immune response have in regeneration is entirely unknown.

The exemplary regenerative capabilities and conservation of multiple immune signaling genes in planarians make this organism an attractive model for understanding how robust tissue regeneration capabilities can be balanced with an effective immune response (*Peiris et al., 2014*). Previous studies in planaria have uncovered components of the immune system conserved in humans, but absent from the well-studied innate immunity models of ecdysozoa (*e.g.*, flies and nematodes) (*Abnave et al., 2014*). This indicates that planaria can serve as a complement to previously established invertebrate models to inform our understanding of the human immune response. While their potent regenerative capabilities and robust capacity for pathogen clearance render them quite resilient, planaria are not invincible. Planaria reared using traditional static culture methods can exhibit features of declining health including decreased appetite, loss of motility, dorsal tissue lesions, tissue degeneration, and lysis. Many of these symptoms can be temporarily alleviated and managed with antibiotics, suggesting that endogenous bacteria can play an antagonistic role in normal

regenerative and tissue homeostatic functions. To date much of the planarian immune system and its potential integration with tissue regenerative processes are still a mystery. Importantly, the composition of the planarian microbiome is still unresolved and no direct link between microbiota, the immune system, and tissue regeneration has been established.

In this study, we investigated the relationship between endogenous microbiota, host response, and regeneration. We performed deep sequencing of bacterial 16 s rDNA under various perturbations to elucidate the composition and dynamics of the planarian microbiome. Importantly, we found that transitions in culture conditions or tissue amputation elicited a robust Proteobacteria expansion that coincided with an increase in the susceptibility of intact worms to develop ectopic tissue lesions. Isolation and identification of bacteria that expanded during this time yielded a strain of *Pseudomonas* capable of inducing progressive tissue degeneration that resulted in either total lysis or resolved to permit regeneration of lost structures. The development of novel low septic culture methods facilitated a candidate RNAi screen examining the mediators of this pathological progression in planaria. We successfully identified a conserved TAK1/MKK/p38 signaling module that both mediated infection-induced tissue degeneration and specifically impeded regeneration during infection. In situ phospho-signaling analyses permitted us to identify tissue and cell types that activate p38 signaling in response to infection, revealing a role in the initiation of localized apoptosis preceding tissue degeneration. This apoptotic and tissue degeneration response coincided with an immune response mediated by *jun D*, an AP-1 family homolog downstream of TAK1 signaling. Finally, while TAK1/MKK/p38 signaling facilitates apoptosis during infection, it plays a contrasting role in the inhibition of apoptosis during normal regeneration. Altogether, our study uncovered potential mechanisms by which immunity and regeneration intersect to mediate distinct tissue regeneration outcomes, and introduces planarians as the first animal model linking the expansion of endogenous Proteobacteria to inhibition of complex tissue regeneration via the activation of distinct TAK1/MKK/p38 signaling.

## Results

### The *S. mediterranea* microbiome is dynamic and responds to changes in culture conditions

To increase the scale and efficiency planarian husbandry, we developed a novel recirculation culture system for co-cultivation of massive numbers of worms. In brief, this system enables the constant recirculation and UV sanitization of planarian water to mitigate pathogenic levels of bacteria (*Figure 1A*). The result is a permissive context for rearing large biomass levels of healthy, fissioning, lesion-free planarians. Adoption of this culture system offered a unique opportunity to study the heretofore unknown composition and dynamics of the planarian microbiome. We hypothesized that bacteria tightly associated with the physiology of healthy worms would be preserved within this recirculation culture while additional, potentially harmful species may accumulate upon exit to traditional static culture conditions. Previously, declining worm health within the static culture was remedied by administration of antibiotic. We assayed parallel cohorts of worms transitioned from recirculation to static culture with or without the antibiotic gentamycin for characterization of the composition and dynamics of the planarian microbiome (*Figure 1A*).

We surveyed the endogenous bacteria of the asexual CIW4 lab strain of *S. mediterranea* via deep sequencing of the 16 s rDNA variable region (*Figure 1—source data 1*). As a control, bacteria present in the beef liver food source of lab-raised planarians was also analyzed but yielded bacterial levels comparable to negative controls and was not considered further. Initial analysis of the planarian microbiome revealed as many as 350 distinct species (*Table 1*), making it more akin to zebrafish than flies in terms of overall complexity (*Lee and Brey, 2013*). The most prevalent bacterial phyla within planaria cultured in the recirculation system was Bacteroidetes (*Figure 1—figure supplement 1A*). This phylum represents a diverse array of key symbionts that support homeostatic functions in mammals (*Rakoff-Nahoum et al., 2004*; *Mazmanian et al., 2005*, *2008*; *Alegado et al., 2012*). Bacteroidetes comprised 78% of the bacterial phyla of planaria from the recirculation system (static culture day 0), but this proportion declined to 57% after 3 days of static culture (*Figure 1B*, *Figure 1—figure supplement 1A*). Administration of gentamycin exacerbated this effect, reducing Bacteroidetes composition to 17%. In the wake of this Bacteroidetes decrease, the proportion of bacteria of the phylum Proteobacteria steadily increased from 9% to 40–44% upon exit from the recirculation

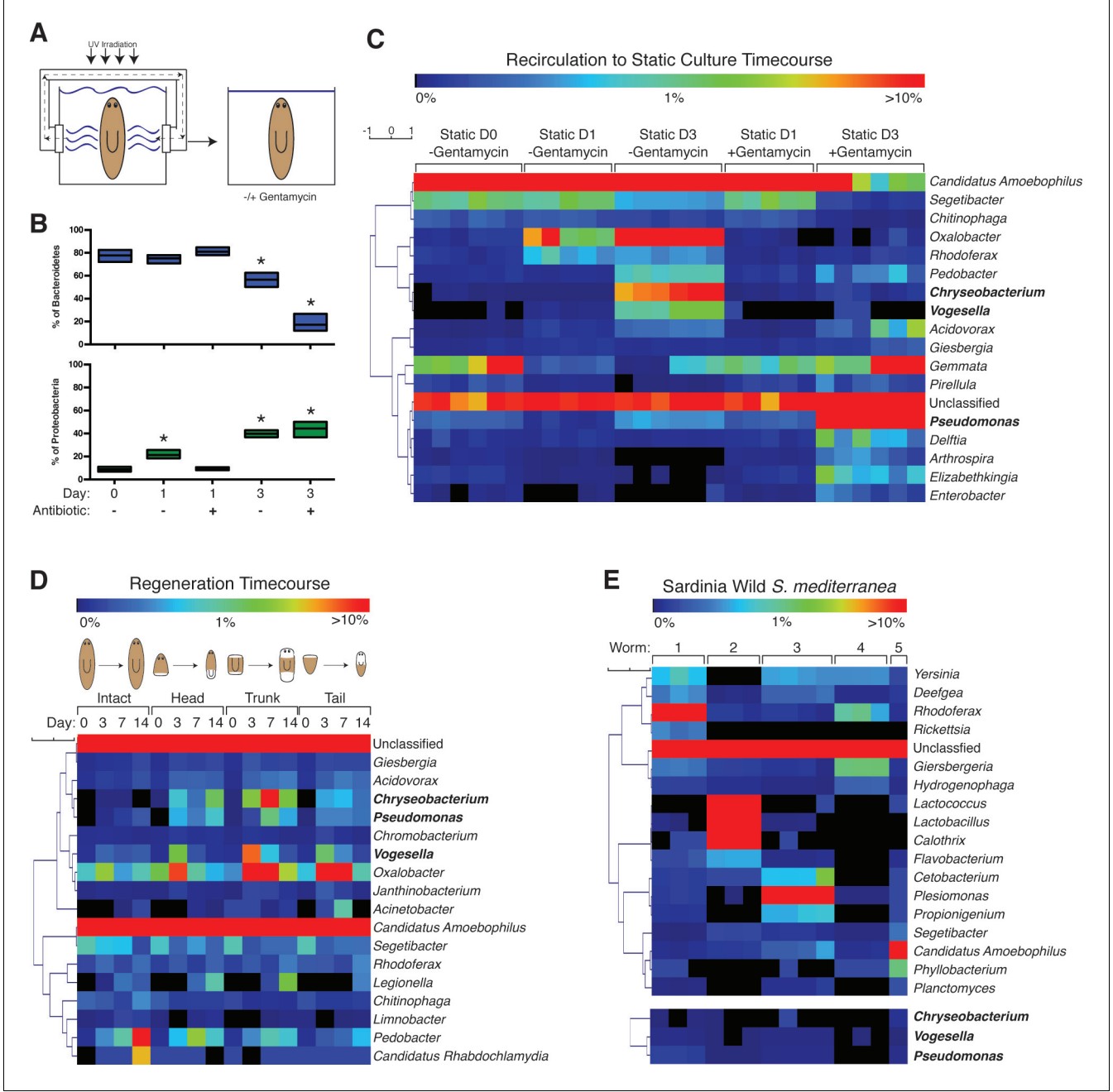

**Figure 1.** The planaria microbiome dynamically responds to changes in culture conditions and regeneration. (**A**) Diagram of transition of planaria from recirculation to static culture. (**B**) Percentage of the phyla Bacteroidetes and Proteobacteria following exit from recirculation culture (* = t-test p<0.05). (**C**) Heatmap of the percentage of the top 10 bacterial genera across all the samples of CIW4 strain planaria following release from recirculation culture in the presence or absence of the antibiotic gentamycin (n = pool of ~30 worms with 2–3 generated 16 s rDNA libraries, independently sequenced twice). (**D**) Heatmap of the percentage of the top 18 bacterial genera of amputated head, trunk, and tail fragments during regeneration in comparison to intact worms (n = pool of 2–16 intact worms or 30–60 fragments for 16 s rDNA library generation). (**E**) Heatmap of the percentage of the top 5 bacterial genera across the samples of individual wild-type planaria. The heatmap of the genera *Vogesella, Chryseobacterium*, and *Pseudomonas* across wild worm samples are included for reference (n = 1–4 16 s rDNA libraries generated per worm).

The following source data and figure supplement are available for figure 1:

**Source data 1.** 16 s rDNA sequencing results.

**Figure supplement 1.** Analysis of changes in bacterial levels and composition following exit from recirculation culture.

**Table 1.** Distribution of Species reads in CIW4 *S. mediterranea.*

| Sample | No. of species (Average) | | | No. of species (Standard deviation) | | |
|---|---|---|---|---|---|---|
| | ≥1 read | ≥10 reads | ≥100 reads | ≥1 read | ≥10 reads | ≥100 reads |
| Recirc D0 | 268.7 | 85.3 | 37.5 | 36.3 | 13.99 | 5.3 |
| D1 -Gent | 245.8 | 79.6 | 34.2 | 17.4 | 5.1 | 0.8 |
| D1 +Gent | 281.8 | 89.4 | 41.2 | 10.8 | 7.8 | 2.7 |
| D3 -Gent | 368 | 144 | 58.8 | 57.2 | 28 | 11.6 |
| D3 +Gent | 354.6 | 126.4 | 57.2 | 26.5 | 14.85 | 5.8 |

system (*Figure 1B*, *Figure 1—figure supplement 1A*). This phylum comprises a wide variety of pathogens and increases in its abundance are a hallmark of microbial dysbiosis and pathological inflammation (*Carvalho et al., 2012*; *Clark et al., 2015*; *Shin et al., 2015*). Thus, transitions in culture conditions are coincident with shifts in bacterial composition with potential implications for planarian health.

We next analyzed compositional changes amongst the most abundant genera of the planarian microbiome following removal from the recirculation system (*Figure 1C*). The most abundant bacterial genera present in recirculation-cultured worms largely belonged to the phylum Bacteroidetes (*Candidatus Amoebophilus, Segetibacter,* and *Chitinophaga*) and declined in relative abundance in static culture (*Figure 1C*, *Figure 1—figure supplement 1B*). The genera *Oxalobacter, Rhodoferax,* and *Vogesella,* belonging to the phylum Proteobacteria, exhibited proportional increases ranging from 6 to 200 fold in worms transferred to static culture (*Figure 1C*, *Figure 1—figure supplement 1C*). Two genera of the phylum Bacteroidetes, *Chryseobacterium* and *Pedobacter*, also increased dramatically after transfer despite the net decrease in the phylum overall. Members of the genus *Chryseobacterium* are potential pathogens in immune-compromised individuals and newborns undergoing prolonged antibiotic treatment (*Bloch et al., 1997*; *Calderón et al., 2011*), suggesting that dramatic shifts in abundance may constitute an opportunistic infection. All of the aforementioned genera were largely gentamycin sensitive. In contrast, proportional increases of *Pseudomonas* in static culture were exacerbated by gentamycin treatment (*Figure 1C*, *Figure 1—figure supplement 1D*), consistent with its adept nature to adapt to the selective pressure of antibiotics (*Hoffman et al., 2005*; *Price et al., 2009*). Additionally, accumulation of *Delftia, Arthrospira, Elizabethkingia,* and *Enterobacter* was largely gentamycin dependent (*Figure 1C*). Collectively, these data suggest that enrichment of a particular cohort of bacterial species may be more conducive to optimal planarian health and destabilization of this microbiota permits amplification of potentially harmful species.

## Regeneration shifts the composition of the *S. mediterranea* microbiome

Planaria exhibit a robust capacity to replace and integrate missing tissues through a regenerative process combining morphogenesis with the remodeling of pre-existing tissues. The extent to which this process impacts the resident microbiome is heretofore unknown. We transferred worms from recirculating into static culture conditions and amputated the animals above and below the pharynx 3 days later. To mitigate bacterial bloom after exit from recirculation culture, worms were starved for 2 weeks prior to transfer and cultured at low density with frequent water changes. Amputated head, trunk, and tail fragments were separated and 16 s sequences were analyzed at 0dpa, 3dpa, 7dpa, and 14dpa. Un-amputated worms were analyzed in parallel as a control (*Figure 1—source data 1*).

Analysis of planarian bacterial phyla during regeneration revealed dramatic shifts in composition that mirrored those seen during the transition from recirculation to static culture (*Figure 1—figure supplement 1A,E*). Prolonged starvation and additional water changes successfully mitigated the Proteobacteria bloom, with Bacteroidetes comprising 75 to 81% of the bacterial phyla of intact worms and freshly amputated worm fragments on day 0. Three days after amputation, worms exhibited a reciprocal decrease of Bacteroidetes and amplification of Proteobacteria phyla (*Figure 1—figure supplement 1E*). In contrast, un-amputated worms cultured in parallel maintained a relative

**Table 2.** 16 s rDNA homology of emergent bacterial strains.

| Planarian source | Colony morphology | Top 10 sequence hits | Max score | Total score | Query cover | E value | Ident | Accession |
|---|---|---|---|---|---|---|---|---|
| Sexual strain S2F2 | Large White | Hafnia paralvei strain ATCC 29927 16 sribosomal RNA gene, partial sequence | 1991 | 1991 | 98% | 0 | 96% | NR_116898.1 |
| | | Obesumbacterium proteus strain 42 16 s ribosomal RNA gene, partial sequence | 1932 | 1932 | 98% | 0 | 96% | NR_025334.1 |
| | | Hafnia alvei strain JCM 1666 16 s ribosomal RNA gene, partial sequence | 1927 | 1927 | 98% | 0 | 96% | NR_112985.1 |
| | | Obesumbacterium proteus strain NCIMB 8771 16 s ribosomal RNA gene, partial sequence | 1927 | 1927 | 98% | 0 | 95% | NR_116603.1 |
| | | Hafnia alvei strain ATCC 13337 16 s ribosomal RNA gene, complete sequence | 1917 | 1917 | 98% | 0 | 95% | NR_044729.2 |
| | | Ewingella americana strain CIP 81.94 16 s ribosomal RNA gene, complete sequence | 1908 | 1908 | 93% | 0 | 97% | NR_104925.1 |
| | | Rouxiella chamberiensis 16 s ribosomal RNA, partial sequence | 1881 | 1881 | 93% | 0 | 97% | NR_135871.1 |
| | | Hafnia psychrotolerans strain DJC1-1 16 s ribosomal RNA, partial sequence | 1875 | 1875 | 93% | 0 | 97% | NR_134741.1 |
| | | Serratia liquefaciens strain ATCC 27592 16 s ribosomal RNA gene, complete sequence | 1871 | 1871 | 98% | 0 | 95% | NR_121703.1 |
| | | Serratia grimesii strain DSM 30063 16 s ribosomal RNA gene, partial sequence | 1871 | 1871 | 98% | 0 | 95% | NR_025340.1 |
| | Medium yellowish beige | Pseudomonas peli strain R-20805 16 s ribosomal RNA gene, partial sequence | 2002 | 2002 | 98% | 0 | 97% | NR_042451.1 |
| | | Pseudomonas guineae strain M8 16 s ribosomal RNA gene, partial sequence | 1953 | 1953 | 98% | 0 | 96% | NR_042607.1 |
| | | Pseudomonas anguilliseptica strain S 1 16 s ribosomal RNA gene, partial sequence | 1908 | 1908 | 98% | 0 | 96% | NR_029319.1 |
| | | Pseudomonas cuatrocienegasensis strain 1N 16 s ribosomal RNA gene, partial sequence | 1897 | 1897 | 98% | 0 | 95% | NR_044569.1 |
| | | Pseudomonas pseudoalcaligenes strain Stanier 63 16 s ribosomal RNA gene, partial sequence | 1868 | 1868 | 98% | 0 | 95% | NR_037000.1 |
| | | Pseudomonas pseudoalcaligenes strain NBRC 14167 16 s ribosomal RNA gene, partial sequence | 1866 | 1866 | 98% | 0 | 95% | NR_113653.1 |
| | | Pseudomonas indoloxydans strain IPL-1 16 s ribosomal RNA gene, partial sequence | 1866 | 1866 | 98% | 0 | 95% | NR_115922.1 |
| | | Pseudomonas alcaligenes strain ATCC 14909 16 s ribosomal RNA gene, partial sequence | 1864 | 1864 | 98% | 0 | 95% | NR_114472.1 |
| | | Pseudomonas alcaligenes strain NBRC 14159 16 s ribosomal RNA gene, partial sequence | 1864 | 1864 | 98% | 0 | 95% | NR_113646.1 |
| | | Pseudomonas composti strain C2 16 s ribosomal RNA gene, partial sequence | 1864 | 1864 | 98% | 0 | 95% | NR_116992.1 |
| | Small Beige with circle | Vogesella mureinivorans strain 389 16 s ribosomal RNA gene, partial sequence | 2039 | 2039 | 99% | 0 | 97% | NR_104556.1 |
| | | Vogesella perlucida strain DS-28 16 s ribosomal RNA gene, partial sequence | 2039 | 2039 | 99% | 0 | 97% | NR_044326.1 |
| | | Vogesella amnigena strain Npb-02 16 s ribosomal RNA, partial sequence | 1967 | 1967 | 99% | 0 | 96% | NR_137334.1 |
| | | Vogesella oryzae strain L3B39 16 s ribosomal RNA gene, partial sequence | 1934 | 1934 | 99% | 0 | 95% | NR_135212.1 |
| | | Vogesella lacus strain GR13 16 s ribosomal RNA gene, partial sequence | 1895 | 1895 | 99% | 0 | 95% | NR_116268.1 |
| | | Vogesella fluminis strain Npb-07 16 s ribosomal RNA gene, partial sequence | 1890 | 1890 | 99% | 0 | 95% | NR_109463.1 |
| | | Vogesella indigofera strain ATCC 19706 16 s ribosomal RNA gene, complete sequence | 1869 | 1869 | 99% | 0 | 94% | NR_040800.1 |

*Table 2 continued on next page*

*Table 2 continued*

| Planarian source | Colony morphology | Top 10 sequence hits | Max score | Total score | Query cover | E value | Ident | Accession |
|---|---|---|---|---|---|---|---|---|
| | | Vogesella alkaliphila strain JC141 16 s ribosomal RNA gene, partial sequence | 1862 | 1862 | 99% | 0 | 94% | NR_108891.1 |
| | | Gulbenkiania mobilis strain E4FC31 16 s ribosomal RNA gene, complete sequence | 1805 | 1805 | 97% | 0 | 94% | NR_042548.1 |
| | | Gulbenkiania indica strain HT27 16 s ribosomal RNA gene, partial sequence | 1762 | 1762 | 97% | 0 | 93% | NR_115769.1 |
| Sexual strain S2F2 | Tiny White | Carbophilus carboxidus strain Z-1171 16 s ribosomal RNA gene, complete sequence | 1951 | 1951 | 98% | 0 | 96% | NR_104931.1 |
| | | Aminobacter aminovorans strain DSM 7048 16 s ribosomal RNA gene, partial sequence | 1951 | 1951 | 98% | 0 | 96% | NR_025301.1 |
| | | Aminobacter lissarensis strain CC495 16 s ribosomal RNA gene, complete sequence | 1945 | 1945 | 98% | 0 | 96% | NR_041724.1 |
| | | Aminobacter niigataensis strain DSM 7050 16 s ribosomal RNA gene, partial sequence | 1940 | 1940 | 98% | 0 | 96% | NR_025302.1 |
| | | Aminobacter aganoensis strain TH-3 16 s ribosomal RNA gene, partial sequence | 1940 | 1940 | 98% | 0 | 96% | NR_028876.1 |
| | | Aminobacter anthyllidis strain STM4645 16 s ribosomal RNA gene, partial sequence | 1934 | 1934 | 98% | 0 | 96% | NR_108530.1 |
| | | Aminobacter ciceronei strain IMB-1 16 s ribosomal RNA gene, complete sequence | 1929 | 1929 | 98% | 0 | 96% | NR_041700.1 |
| | | Mesorhizobium australicum strain WSM2073 16 s ribosomal RNA gene, complete sequence | 1882 | 1882 | 94% | 0 | 96% | NR_102452.1 |
| | | Mesorhizobium qingshengii strain CCBAU 33460 16 s ribosomal RNA gene, partial sequence | 1882 | 1882 | 94% | 0 | 96% | NR_109565.1 |
| | | Mesorhizobium shangrilense strain CCBAU 65327 16 s ribosomal RNA gene, partial sequence | 1882 | 1882 | 94% | 0 | 96% | NR_116163.1 |
| Asexual strain CIW4 | Small Beige with circle | Vogesella perlucida strain DS-28 16 s ribosomal RNA gene, partial sequence | 2167 | 2167 | 94% | 0 | 97% | NR_044326.1 |
| | | Vogesella mureinivorans strain 389 16 s ribosomal RNA gene, partial sequence | 2156 | 2156 | 94% | 0 | 97% | NR_104556.1 |
| | | Vogesella lacus strain GR13 16 s ribosomal RNA gene, partial sequence | 2019 | 2019 | 95% | 0 | 95% | NR_116268.1 |
| | | Vogesella oryzae strain L3B39 16 s ribosomal RNA gene, partial sequence | 2013 | 2013 | 95% | 0 | 95% | NR_135212.1 |
| | | Vogesella fluminis strain Npb-07 16 s ribosomal RNA gene, partial sequence | 2012 | 2012 | 94% | 0 | 95% | NR_109463.1 |
| | | Vogesella indigofera strain ATCC 19706 16 s ribosomal RNA gene, complete sequence | 1989 | 1989 | 94% | 0 | 94% | NR_040800.1 |
| | | Vogesella alkaliphila strain JC141 16 s ribosomal RNA gene, partial sequence | 1980 | 1980 | 94% | 0 | 94% | NR_108891.1 |
| | | Gulbenkiania mobilis strain E4FC31 16 s ribosomal RNA gene, complete sequence | 1884 | 1884 | 94% | 0 | 93% | NR_042548.1 |
| | | Pseudogulbenkiania gefcensis strain yH16 16 s ribosomal RNA gene, partial sequence | 1857 | 1857 | 95% | 0 | 92% | NR_118145.1 |
| | | Aquaphilus dolomiae strain LMB64 16 s ribosomal RNA gene, partial sequence | 1855 | 1855 | 94% | 0 | 93% | NR_118538.1 |
| | Medium Yellow | Chryseobacterium lactis strain KC1864 16 s ribosomal RNA gene, partial sequence | 2017 | 2017 | 98% | 0 | 96% | NR_126256.1 |
| | | Chryseobacterium viscerum strain 687B-08 16 s ribosomal RNA gene, partial sequence | 2012 | 2012 | 98% | 0 | 95% | NR_117206.1 |
| | | Chryseobacterium tructae strain 1084-08 16 s ribosomal RNA gene, partial sequence | 1995 | 1995 | 98% | 0 | 95% | NR_108531.1 |
| | | Chryseobacterium oncorhynchi strain 701B-08 16 s ribosomal | 1995 | 1995 | 98% | 0 | 95% | NR_108481.1 |

*Table 2 continued on next page*

*Table 2 continued*

| Planarian source | Colony morphology | Top 10 sequence hits | Max score | Total score | Query cover | E value | Ident | Accession |
|---|---|---|---|---|---|---|---|---|
| | | RNA gene, partial sequence | | | | | | |
| | | Chryseobacterium ureilyticum strain F-Fue-04IIIaaaa 16 s ribosomal RNA gene, partial sequence | 1980 | 1980 | 98% | 0 | 95% | NR_042503.1 |
| | | Chryseobacterium indologenes strain NBRC 14944 16 s ribosomal RNA gene, partial sequence | 1969 | 1969 | 98% | 0 | 95% | NR_112975.1 |
| | | Chryseobacterium gleum strain NBRC 15054 16 s ribosomal RNA gene, partial sequence | 1967 | 1967 | 98% | 0 | 95% | NR_113722.1 |
| | | Chryseobacterium gleum strain CCUG 14555 16 s ribosomal RNA gene, partial sequence | 1967 | 1967 | 98% | 0 | 95% | NR_042506.1 |
| | | Chryseobacterium indologenes strain LMG 8337 16 s ribosomal RNA gene, partial sequence | 1964 | 1964 | 98% | 0 | 95% | NR_042507.1 |
| | | Chryseobacterium artocarpi strain UTM-3 16 s ribosomal RNA, partial sequence | 1960 | 1960 | 98% | 0 | 95% | NR_134001.1 |
| Asexual strain CIW4 | Large Beige | Pseudomonas fluorescens Pf0-1 strain Pf0-1 16 s ribosomal RNA, complete sequence | 2242 | 2242 | 94% | 0 | 99% | NR_102835.1 |
| | | Pseudomonas koreensis strain Ps 9-14 16 s ribosomal RNA gene, partial sequence | 2220 | 2220 | 94% | 0 | 98% | NR_025228.1 |
| | | Pseudomonas reinekei strain MT1 16 s ribosomal RNA gene, partial sequence | 2217 | 2217 | 94% | 0 | 98% | NR_042541.1 |
| | | Pseudomonas moraviensis strain 1B4 16 s ribosomal RNA gene, partial sequence | 2215 | 2215 | 94% | 0 | 98% | NR_043314.1 |
| | | Pseudomonas vancouverensis strain DhA-51 16 s ribosomal RNA gene, partial sequence | 2215 | 2215 | 94% | 0 | 98% | NR_041953.1 |
| | | Pseudomonas helmanticensis strain OHA11 16 s ribosomal RNA gene, partial sequence | 2193 | 2193 | 94% | 0 | 98% | NR_126220.1 |
| | | Pseudomonas baetica strain a390 16 s ribosomal RNA gene, partial sequence | 2193 | 2193 | 94% | 0 | 98% | NR_116899.1 |
| | | Pseudomonas jessenii strain CIP 105274 16 s ribosomal RNA gene, partial sequence | 2185 | 2185 | 94% | 0 | 98% | NR_024918.1 |
| | | Pseudomonas umsongensis strain Ps 3-10 16 s ribosomal RNA gene, partial sequence | 2170 | 2170 | 94% | 0 | 98% | NR_025227.1 |
| | | Pseudomonas mucidolens strain NBRC 103159 16 s ribosomal RNA gene, partial sequence | 2167 | 2167 | 94% | 0 | 98% | NR_114225.1 |

abundance of Bacteroidetes until the final 2-week time point. This eventual shift is likely the result of prolonged static culture conditions. Next, we analyzed proportional changes in the top genera of intact and regenerating worms. The genera *Candidatus Amoebophilus* and *Segetibacter*, both members of the phylum Bacteroidetes, decreased in abundance 3 days after amputation (*Figure 1D*, *Figure 1—figure supplement 1F*). In parallel, the genera *Acidovorax*, *Chryseobacterium*, *Pseudomonas*, *Vogesella*, and *Oxalobacter* exhibited transient or sustained increases in regenerating worm fragments 3 to 7 days after amputation (*Figure 1D*, *Figure 1—figure supplement 1G,H*). Thus, the expansion of Proteobacteria observed in both regeneration and static culture time courses can be traced to common genera amplified under both conditions.

## The microbiomes of wild and lab-raised planaria share many genera

To determine how the bacterial composition of the asexual CIW4 lab strain compared to that of wild planaria, we performed 16 s sequencing on individual sexual *S. mediterranea* samples collected from multiple sites in Sardinia. Negative controls were performed in tandem during sample and library preparation to exclude contamination from resident lab strains. We succeeded in amplifying and sequencing 16 s libraries from 5 out of 8 worms covering 2 collection sites (*Figure 1—source data 1*). Phyla composition varied between individual animals. Proteobacteria was either the most

abundant (3 out of 5 worms) or the second most abundant (2 out of 5 worms) phyla across all worm samples (*Figure 1—figure supplement 1I*). Bacteroidetes or Firmicutes were the dominant phyla in the two cases in which Proteobacteria was not the most abundant. These differences in phyla composition were also reflected in the most abundant bacterial genera across individuals. The genera in the highest proportion was *Rhodoferax*, *Lactococcus*, *Plesiomonas*, *Giesbergeria*, and *Candidatus amoebophilus* in worms 1, 2, 3, 4, and 5, respectively (*Figure 1E*). This level of variation in endogenous bacterial composition amongst individuals is consistent with previous observations in other animals (*Caporaso et al., 2011*; *Human Microbiome Project Consortium, 2012*; *Wong et al., 2013*). Furthermore, the phylum Proteobacteria was proportionally higher in wild collected versus lab-raised worms consistent with observations flies (*Wong et al., 2013*). Interestingly, we found that the genera that increased during transitions in culture conditions and regeneration, (*Chryseobacterium*, *Vogesella*, and *Pseudomonas*), were detectable in wild planaria samples but present at relatively low levels (*Figure 1E*, *Figure 1—figure supplement 1J*).

Despite this heterogeneity in bacterial composition, more bacterial genera were common across all four worms collected from site#1 in Sardinia than were unique to any individual (*Figure 1—figure supplement 1K*). Furthermore, greater than 90% of the bacterial genera detected among all of our CIW4 lab strain planaria samples were also detected among our wild worm samples (*Figure 1—figure supplement 1L*). With respect to wild worms, this level of overlap was ~46%, suggesting that a significant proportion of bacterial genera is distinct in our wild sexual samples. To determine possible differences between asexual versus sexual biotypes, we sampled prominent culturable bacteria from lab raised sexual planaria. We utilized sexual S2F2 genome strain planaria from a fill and drain culture system that were transitioned to traditional static culture. We detected abundant *Vogesella*, *Pseudomonas*, *Aminobacter*, and *Hafnia* bacterial colonies from sexual worms at day 3 of static culture (*Figure 1—figure supplement 1M*, *Table 2*). This limited sampling demonstrates that asexual and sexual lab cultured planaria have both similar and distinct bacterial composition. Therefore, with respect to wild sexual bacteria, it is unclear to what extent the observed differences with asexual lab planaria are attributable to (1) intrinsic microbiome differences between the asexual and sexual biotypes, (2) region-specific bacterial genera, and/or (3) shifts in genera during lab culture. Importantly, our analyses demonstrate that the vast majority of genera from CIW4 lab strain planaria are common to those of wild type sexual worms.

## Bacterial blooms observed after removal from recirculation culture or amputation are due to specific bacterial strains

We quantitated endogenous planarian bacterial levels by either 16 s rDNA qPCR or plating tissue homogenates onto LB plates. Removal of animals from the recirculation culture system elicited a 17-fold expansion in 16 s rDNA over three days that was largely mitigated by gentamycin treatment (*Figure 1—figure supplement 1N*). Additionally, both removal from the recirculation culture system and amputation elicited a 2- to 3-log fold increase in score-able bacterial species per worm within 3 days (*Figure 1—figure supplement 1O,P*). Representative colony 16 srDNA amplification and sequencing revealed that the observed bacterial bloom was dominated by three bacterial species belonging to the following genera: *Vogesella*, *Chryseobacterium*, and *Pseudomonas* (*Figure 2A*, *Table 2*). The expansion of these bacteria during culture transition was highly reproducible in multiple withdrawals from two independent recirculation culture systems (data not shown). These findings are consistent with our 16 s rDNA sequencing data showing proportional increases of these genera within worms upon exit from recirculation culture (*Figure 1C*, *Figure 1—figure supplement 1C,D*). Thus, shifts in microbial composition are coincident with a robust expansion in both absolute bacterial levels and particular bacterial species.

## Infection of planaria with emergent bacterial strains yields progressive tissue degeneration

Following the transition from recirculation to static culture, poorly managed worms are highly susceptible to the development of dorsal lesions, tissue degeneration, and lysis. This susceptibility can be exacerbated by increased feedings or temporarily mitigated by antibiotics (*Figure 2—figure supplement 1A*). We hypothesized the etiology of this tissue degeneration lie in the observed bacterial bloom upon exit from recirculation system (*Figures 1C*, *2A*, *Figure 1—figure supplement 1C,D*).

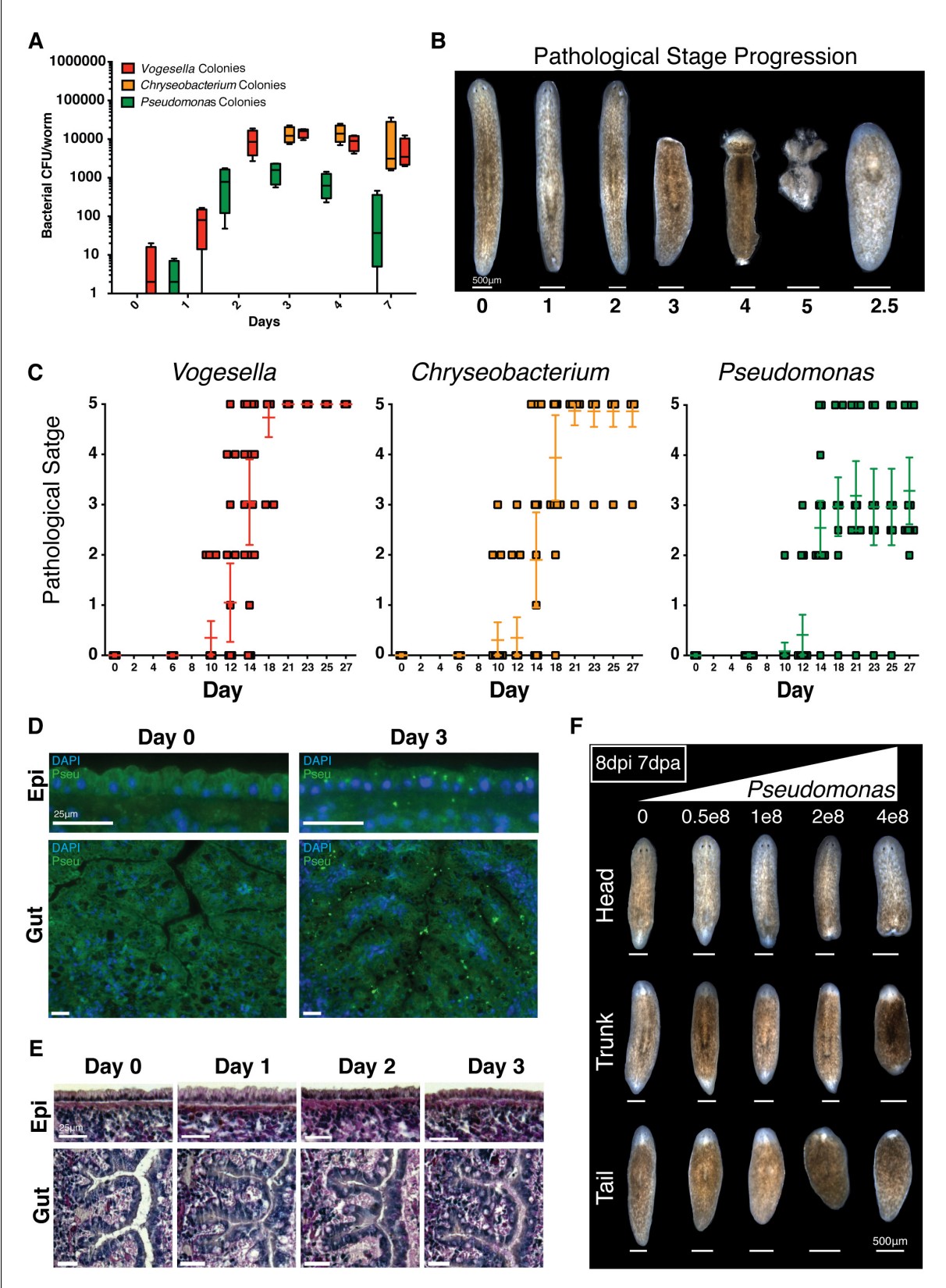

**Figure 2.** Bacterial infection compromises planarian tissue homeostasis and regeneration. (**A**) Bacterial CFU quantification of prevalent bacterial strains following exit from the recirculation system (n = 4 homogenates from 4 worms per time point, experiment independently repeated > 4 times); *Figure 2 continued on next page*

*Figure 2 continued*

*Vogesella* (red), *Chryseobacterium* (orange), and *Pseudomonas* (green). (B) Depiction and enumeration of pathological stages following bacterial infection. Stage 2.5 refers to worms that reach stage 3 but ultimately regenerated anterior structures. (C) Comparison of the effects of infection of 1e8 CFU/ml *Vogesella*, *Chryseobacterium*, or *Pseudomonas* (green) on pathological stage progression over time (n = 15–25, experiment independently repeated > 3 times). (D) Anti-*Pseudomonas* antibody staining (green) and DAPI nuclear counterstain (blue) of surface and gut epithelia following infection (n = 2, experiment independently repeated > 2 times). (E) Histological sections stained with Alcian Blue/ PAS following *Pseudomonas* infection (n = 2). (F) Representative images depicting the effects of increasing concentrations of *Pseudomonas* on regenerating head, trunk, and tail fragments. Worms were amputated 1 day following infection and images were taken seven days after amputation (n = 5).

The following figure supplement is available for figure 2:

**Figure supplement 1.** Effects of bacterial infection on worm tissue homeostasis.

To determine whether this bacteria was sufficient to induce tissue homeostatic defects we introduced emergent *Vogesella*, *Chryseobacterium*, or *Pseudomonas* bacterial strains to planaria water immediately upon transition to static culture. Worms were infected at a concentration of 1E8 CFU/ml and washed every 3–4 days for administration of fresh planaria water containing bacteria. This method of infection lead to a dose dependent increase in endogenous bacterial levels over time (*Figure 2—figure supplement 1B*) similar to those observed in the worms following removal from the recirculation system (*Figure 2A*). In comparison feeding of bacteria mixed with beef liver paste elicited an initial spike and steady decline in bacterial levels as previously reported (*Abnave et al., 2014*), but the endogenous bacterial levels of control fed worms also increased to match that of bacteria fed worms (*Figure 2—figure supplement 1C*). This expansion of endogenous bacteria complicates analysis and obfuscates effects of tested bacterial strains on host biology. We therefore utilized supplementation of bacteria directly in planaria water to determine effects on tissue homeostasis.

In response to infection, worms exhibited pathological defects largely resembling planaria of declining health in static culture (*Figure 2—figure supplement 1A*). We categorized the observed pathologies into the following discrete categories: normal, posterior lesions, anterior lesions, head regression, partial lysis, and full lysis. Infected worms exhibited a consistent development of tissue defects over time varying in the rate of progression by bacterial strain (*Figure 2—figure supplement 1D*). Generally phenotypically normal worms initially developed either posterior or anterior dorsal lesions, anterior lesions increased in severity to result in full head regression, and worms exhibiting head regression ultimately initiated full tissue lysis. The notable exception was the incidence of anterior blastema formation and regeneration as opposed to lysis in a subset of *Pseudomonas* infected worms with regressed heads. These data demonstrate that both the rate of pathological progression and eventual outcome of the infection in planaria varies by bacterial strain.

We were able to infer a chronological hierarchy and assigned specific stages to the observed progressive tissue degeneration: Normal = 0, Posterior Lesion = 1, Anterior Lesion = 2, Head regression = 3, Partial Lysis = 4, Full Lysis = 5 (*Figure 2B*). Worms that regenerated lost anterior structures were assigned stage 2.5, reflecting tissue restoration following stage 3 head regression (*Figure 2B*). Using these stages we plotted tissue degeneration over time, highlighting differential effects of tested strains on planarian tissue homeostasis (*Figure 2C*). We chose *Pseudomonas* for subsequent analyses based on the following criteria: it offered the opportunity to study loss and regeneration of tissues during infection, it is the most well studied of the tested strains, and members of this genus are relevant human pathogens (*Driscoll et al., 2007*).

## *Pseudomonas* localizes to barrier epithelia during the initial stages of infection

We used a validated anti-*Pseudomonas* antibody and histological sections in order to determine tissues likely affected by infection (*Figure 2—figure supplement 1E*). Initially, *Pseudomonas* mainly localized to the barrier epithelia of the worm in the epidermis and intestine (*Figure 2D*). By 3 days some *Pseudomonas* appear to have crossed epithelial barriers and localize to the mesenchyme. In order to study the effects on the mucosal epithelia, we performed Alcian Blue/ Periodic Acid-Schiff staining of histological sections following infection (*Figure 2E*). Infected worms exhibited a gradual

apical compression of the morphology of outer epithelial cells with a slight distortion in the organization of the basement membrane. In the intestine, *Pseudomonas* infection resulted in a pronounced lumenal constriction with changes in epithelial organization and morphology over time. Thus, our data indicate that barrier epithelia represent the initial point of interaction with *Pseudomonas* and that sustained exposure to this bacterium results in observable cellular and tissue morphology changes.

## *Pseudomonas* infection compromises tissue regeneration

Given that *Pseudomonas* infection progressively compromised tissue homeostasis (*Figure 2C,E*), we next determined whether neoblasts and regeneration potential were similarly affected. We analyzed neoblast levels and distribution using WISH analysis. While infected worms exhibited abundant numbers of *piwi*+ neoblasts both prior to and following tissue degeneration, the frequency of neoblasts within the interior of the animal relative to the periphery appeared to progressively decrease during infection (*Figure 2—figure supplement 1F*). To determine how bacterial challenge altered regeneration potential, we infected worms with increasing dosages of *Pseudomonas* and amputated above and below the pharynx one day after exposure to bacteria. Phenotypic effects on resulting head, trunk, and tail fragments were documented one week later. *Pseudomonas* infection inhibited tissue regeneration of amputated fragments in a dose dependent manner (*Figure 2F*). Interestingly, we observed that segments of the worm along the A/P axis had different tolerances for bacterial infection. Head fragments were the most susceptible, tail fragments exhibited intermediate sensitivity, and trunk fragments were the most resistant. These data suggest that differences in tissue composition along the A/P axis and/or the extent of tissue generation versus remodeling required for re-establishment of the body plan confer resistance or susceptibility to infection. Given its effects on tissue regeneration, we chose the dosage of 2e8 CFU/ml *Pseudomonas* for subsequent experiments.

## Induction of innate immunity genes coincides with Proteobacteria expansion

To identify the host transcriptional changes underlying shifts in planarian microbial composition, we conducted RNAseq analyses on worms transferred from recirculation to static culture in the presence or absence of the antibiotic gentamycin (*Figure 3A*, *Figure 3—source data 1*). This transition results in an increase in absolute bacterial levels characterized by a relative decrease in the phyla Bacteroidetes, a proportional increase in the phyla Proteobacteria, and the emergence of newly dominant bacterial genera and species (*Figures 1B,C,2A*, *Figure 1—figure supplement 1N,O*). Clustering of the 1,218 genes significantly altered relative to static culture at day 0 revealed transcriptional patterns that were either dependent (clusters 1, 2, 4, 5, 7, 8,and 9) or independent (clusters 3 and 6) of antibiotic treatment (*Figure 3B,C*, *Figure 3—source data 1*). Gene clusters sensitive to antibiotic treatment could be divided based on up-regulation (clusters 1, 4, 8, 9) or down-regulation (clusters 2, 5, 7), and then further subdivided into phases of early (clusters 4 and 7), mid (cluster 8), late (clusters 1 and 5), or sustained (clusters 2 and 9) expression changes. Of particular interest was cluster 1, consisting of genes upregulated gradually after exit from the recirculation system in the absence of antibiotic (*Figure 3D*). This cluster had clear potential enrichment in genes that correlated with increased bacterial levels over time. Interestingly, this gene cohort contained two transcripts with homology to peptidoglycan recognition proteins (pgrps), upstream receptors of the IMD pathway in *D. melanogaster.* Along with Toll signaling, the IMD pathway represents a conserved branch of the innate immune system with analogous architecture to the inflammatory signaling pathways in vertebrates (*Panayidou and Apidianakis, 2013*). RT/qPCR and WISH analysis confirmed that *Pseudomonas* infection significantly induced one of these homologous receptors, that we designated *pgrp-4* (*Figure 3E–G*). These data suggest that conserved components of the innate immune and inflammatory signaling pathways may dynamically respond to shifts in the microbial composition in planaria. This data as well as the established role of this pathway in immunity and apoptosis lead us to take a candidate based investigation of the role of inflammatory signaling pathways in altered tissue homeostasis and regeneration during infection.

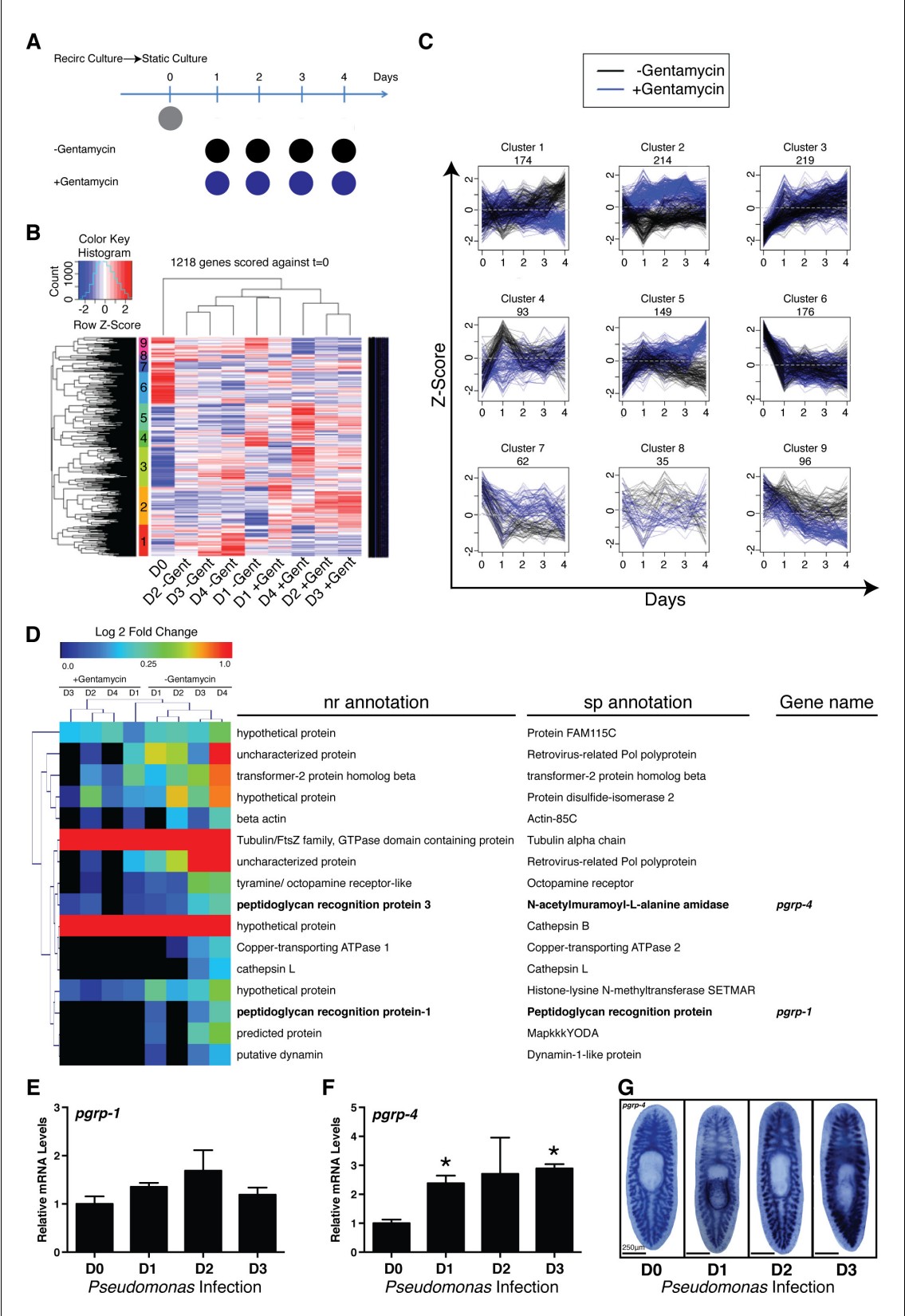

**Figure 3.** Elucidating bacterial contribution to the transcriptional changes underlying the transition of worms from recirculation to static culture conditions. (A) Diagram depicting comparison of RNAseq samples of planaria following exit from a recirculation culture system in the absence or *Figure 3 continued on next page*

*Figure 3 continued*

presence of the antibiotic gentamycin (n = 4 replicates each containing 4 planaria). (B) Hierarchical clustering of RNAseq results displaying significant clusters (1 through 9, y axis). (C) Visualization of gene expression patterns of significant clusters. (D) Differentially expressed annotated genes from cluster 1 (Day 0 vs Day 4 adj. p-value <0.05). Peptidoglycan recognition protein genes are highlighted in bold. RT and QPCR of (E) *pgrp-1* and (F) *pgrp-4* following *Pseudomonas* infection (n = 3 biological replicates, 3 technical replicates, * = t-test p<0.05). (G) WISH of *pgrp-4* following *Pseudomonas* infection (n = 5–7).

The following source data is available for figure 3:

**Source data 1.** RNAseq analysis of worms during recirculation to static culture transition.

## Development of a reduced septic planaria RNAi protocol

The identification of genes modulated by a bacterial infection as well as the appearance of a visibly score-able progression of tissue degeneration presented an opportunity to elucidate genes mediating this process via RNAi-mediated genetic interference. Given their low levels of Proteobacteria and in particular the emergent pathogenic strains we identified (*Figures 1B, C*, *2A*), planaria from the recirculation culture system presented the greatest opportunity for studying the maximal differential effect caused by *Pseudomonas* infection in terms of phenotypic and underlying molecular responses. However, in order to use these animals and conditions for an RNAi screen to identify immune response mediators, we needed to devise a way to reduce the bacterial bloom observed after removal from recirculation culture without using antibiotics (*Figure 4—figure supplement 1A*). As we observed in our experiments, the antibiotic gentamycin substantially altered endogenous microbiota composition, diminishing the phylum Bacteroidetes while enriching for Proteobacteria, including *Pseudomonas* itself (*Figure 1B,C*). Its usage would preclude analysis of phenotypes that are dependent on dynamic shifts in endogenous microbiota. Additionally, a shared recirculation culture system precludes the delivery of multiple, distinct dsRNAs to separate experimental cohorts of planaria.

To overcome these limitations, we developed a unidirectional flow system in which worms were maintained in multiple individual containers and steadily flushed with fresh planaria water (*Figure 4—figure supplement 1A–C*). Culture of planaria under a steady flow rate prevented a 4-log increase in culturable bacteria levels observed in static conditions following exit from the recirculation culture system (*Figure 4—figure supplement 1D*). Additionally, introduction of static worms to flow vessels resulted in a nearly 2 log reduction in culturable bacterial levels over 5 days (*Figure 4—figure supplement 1E*), and worms administered 3 feedings of control dsRNA mixed with beef liver paste maintained levels comparable to those observed in recirculation culture (*Figure 4—figure supplement 1F*). We conclude from these findings that the unidirectional flow system provides the necessary conditions to systematically carry out an RNAi screen of infection-modulated genes and to effectively analyze their resulting phenotypes (*Figure 4—figure supplement 1A*).

## RNAi screen uncovers mediators of tissue degeneration in response to infection

We carried out a candidate gene RNAi screen for mediators of planarian tissue degeneration in response to infection by focusing on conserved members of the mammalian inflammatory signaling cascade and the analogous, infection-responsive IMD pathway in *Drosophila melanogaster*. We identified 32 homologous genes using reciprocal BLAST of either *Homo sapiens* or *D. melanogaster* reference genes and categorized them with respect to known functions in mediating (activators) or reducing (inhibitors) signal transduction (*Figure 5—source data 1*) (*Kopp et al., 1999*; *Mogensen, 2009*; *Dai et al., 2012*; *Xie, 2013*; *Fernando et al., 2014*; *Herrington and Nibbs, 2016*). Additional genes with roles in regulating apoptosis, *bcl2-1* and *bcl2-2,* and one TIR domain containing gene, *tehao*, were also included in our analysis (*Luo et al., 2001*; *Pellettieri et al., 2010*). The RNAi screen was performed in low septic conditions outside of the recirculation culture as follows (*Figure 4*). Worms in individual flow vessels were administered 3 dsRNA feedings. Four days after RNAi treatment, worms were infected with *Pseudomonas* and ocularly scored for pathological progression using established criteria (*Figure 2B*) every 1–3 days over a period of 30 days. Scoring resulted in a series of 5 to 6 integers representing the state of each worm in the dish for each RNAi condition for

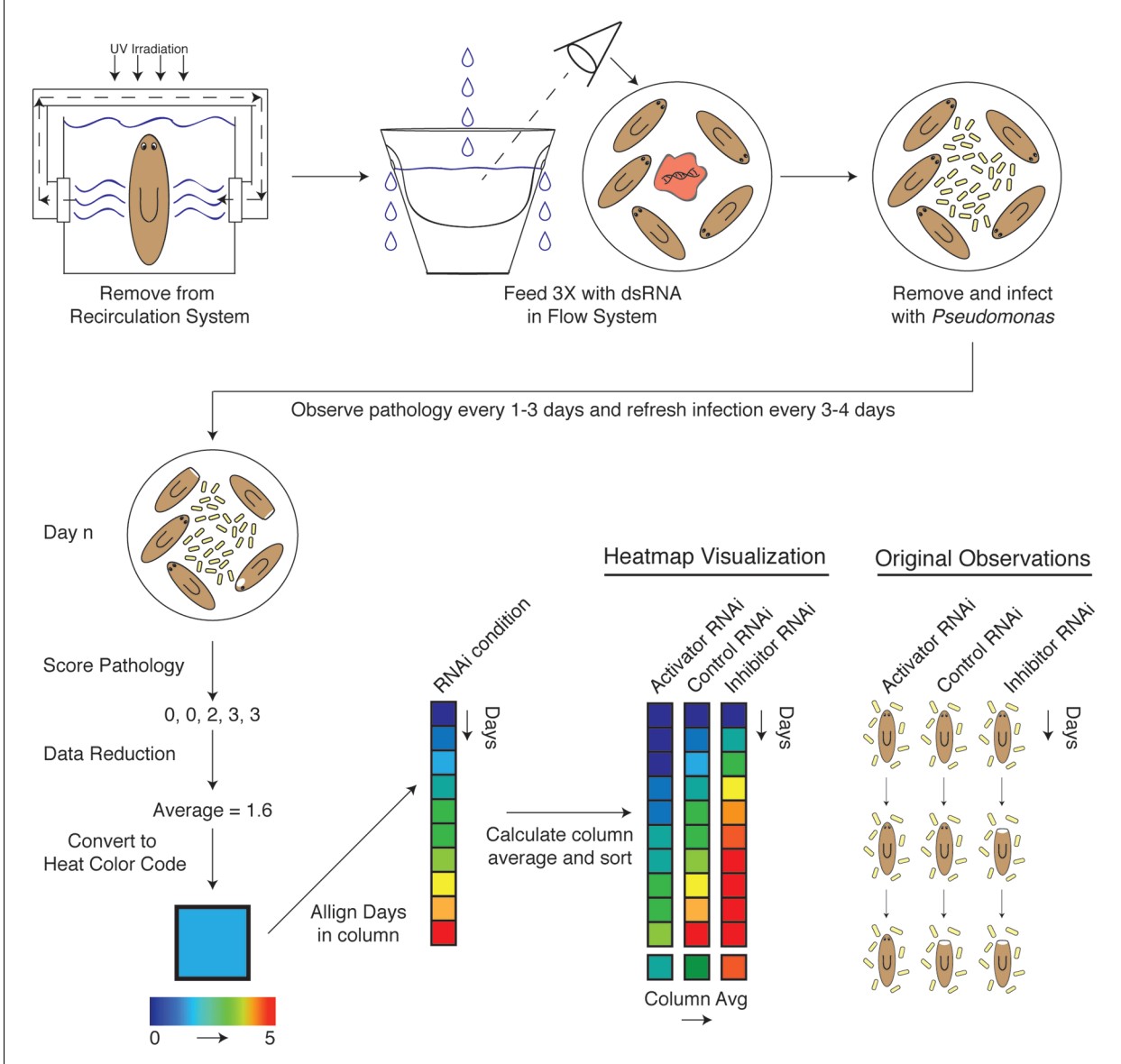

**Figure 4.** Diagram depicting workflow and data analysis of an RNAi screen to identify genes modulating pathological progression. Planaria are transferred directly from the recirculation system to a flow culture system permitting maintenance of a reduced septic state during dsRNA feedings. Following 3 RNAi feedings, planaria are removed for infection with *Pseudomonas*. Observations of pathological stages are recorded every 1 to 3 days, and planaria are replenished with fresh water containing *Pseudomonas* every 3 to 4 days. Following enumeration of pathological stages, data for each day is reduced to an average pathological score and converted to a heat color code. Days for each RNAi condition are aligned in ascending order along the y-axis of a column. The average score of each column is calculated and used to sort the effects of RNAi conditions in ascending order along the x-axis. The result is a heat map visualization ranking the effects of RNAi treatments on the pathological progression in response to bacterial infection in planarians over time.

The following figure supplement is available for figure 4:

**Figure supplement 1.** A novel flow culture method for planarian RNAi in low septic conditions.

each day. By taking the average score of the worms on each day and converting that average to a color-based scale from blue to red to denote pathological severity, we were able to reduce the complexity of the data to facilitate its visualization. For each RNAi condition, color-coded average scores for each day were stacked along the y-axis in ascending order from top to bottom forming a column.

Individual RNAi columns were sorted in ascending order from left to right along the x-axis by their average column scores. The result was a heatmap that ranked the pathological progression of the worms following RNAi treatment in order of increasing overall severity over time. Simply stated, genes in which RNAi treatment reduces pathological progression (presumptive activators) reside left of the control, while genes in which RNAi treatment enhances pathological progression (presumptive inhibitors) reside right of the control.

A total of 20/35 tested candidate genes resulted in a significant change (2-WAY ANOVA $p<0.05$) in pathological progression following infection relative to *unc* controls, while 2/35 genes, *bcl2-1* and *bcl3-1* aka *NFκB-p105*, exhibited total lysis prior to infection and were not considered further (*Figure 5A*) (*Pellettieri et al., 2010*; *Forsthoefel et al., 2012*). Among these, 15/20 significantly reduced pathological progression while 5/20 significantly enhanced it; we termed these genes activators and inhibitors, respectively. Interestingly, the homologs of 13/15 activators (*mkk6-1*, *mkk4*, *p38-1*, *pgrp-2*, *tab1-1*, *jun D*, *hep*, *tak1*, *jnk*, *pgrp-3*, *pgrp-1*, *xiap*, *pgrp-1*) have been previously shown to be positive regulators of inflammatory or innate immune signaling (*Figure 5B*, *Figure 5— source data 2*). Conversely, the homologs for 4/5 inhibitors (*pp6*, *ppm1b*, *cyld-1*, *ppm1a*) have been previously shown to inhibit inflammatory and innate immune signaling (*Figure 5B*, *Figure 5—source data 2*). In the case of the inhibitor *traf2-1*, the homolog *traf2* functions as mediator of inflammatory signaling, but it appears that the function of this gene in planaria is more akin to *traf3*, which functions as an inhibitor (*Xie, 2013*). Overall the data suggest that conserved inflammatory signaling may be largely responsible for mediating tissue degeneration after bacterial infection.

Our data set also permitted us to track the progression of worms through individual pathological stages in response to infection. We mapped the percentage of worms that remained lesion free (stage 0), exhibited head regression (stage 3), fully lysed (stage 5), or regenerated their heads (stage 2.5) onto our average pathological score heatmap (*Figure 5C–F*). Control worms remained lesion free for up to 12 days after infection (*Figure 5C*). RNAi of activators could extend this period up to 20 days. Conversely, RNAi of inhibitors reduced this lesion free period to 6 days. With respect to head regression (stage 3), RNAi of activators largely delayed the initiation of and the extended the duration of this stage while RNAi of inhibitors expedited and contracted it (*Figure 5D*). Following head regression, control worms began undergoing lysis by day 12, and 100% of worms lysed 27 days following infection (*Figure 5E*). RNAi of activators delayed and reduced the incidence of lysis and 100% of *mkk6-1* and *mkk4* RNAi worms survived the entire 30-day duration of our assay. RNAi of inhibitors induced lysis as early as day 4 with no worms surviving after 9 to 23 days. While the vast majority of worms eventually lysed with the administered dosage of 2e8 CFU/ml *Pseudomonas*, a rare fraction of control worms regenerated their heads by day 23 (*Figure 5F*). While RNAi of many activators increased the incidence, frequency, and duration of head regeneration and subsequent survival during infection, we observed no incidence of head regeneration following RNAi of the inhibitors *pp6*, *ppm1b*, *cyld-1* or *traf2-1*. Amongst activators, *mkk4* and *mkk6* RNAi worms had the highest incidence of head regeneration while *p38-1* RNAi had the lowest despite having a comparable frequency and timing of the initiation of head regression (*Figure 5D*). In summation, our RNAi screen has identified genes that play a role in mediating tissue degeneration in response to infection and reveal distinct effects on the timing and duration of individual stages of pathological progression.

## Combinatorial RNAi uncovers hierarchy of genes mediating tissue degeneration during bacterial infection in planarians

Our candidate RNAi screen identified cohorts of genes that serve as activators or inhibitors of the tissue degeneration response to *Pseudomonas* infection. These opposing phenotypes provided an opportunity to order these components with respect to their hierarchy in mediating anterior tissue degeneration. We chose 6 activator genes (*mkk6-1*, *mkk4*, *p38-1*, *jun D*, *tak1*, and *jnk*) and 3 inhibitor genes (*ppm1b*, *pp6*, *cyld-1*) analogous to core inflammatory and innate immune signaling in *H. sapiens* (*Figure 6A*) (*Xue et al., 2007*; *Mogensen, 2009*; *Dai et al., 2012*; *Panayidou and Apidianakis, 2013*). To order these components we performed combinatorial RNAi experiments and assayed the resulting effects on anterior tissue degeneration. The dsRNA concentration of the inhibitors was reduced by 50% and balanced with a 50% increase in the concentration of activators. This allowed us to determine whether simultaneous knock down of activator genes was sufficient to overcome the sensitized tissue degeneration phenotype from partial knockdown of inhibitors. Worms

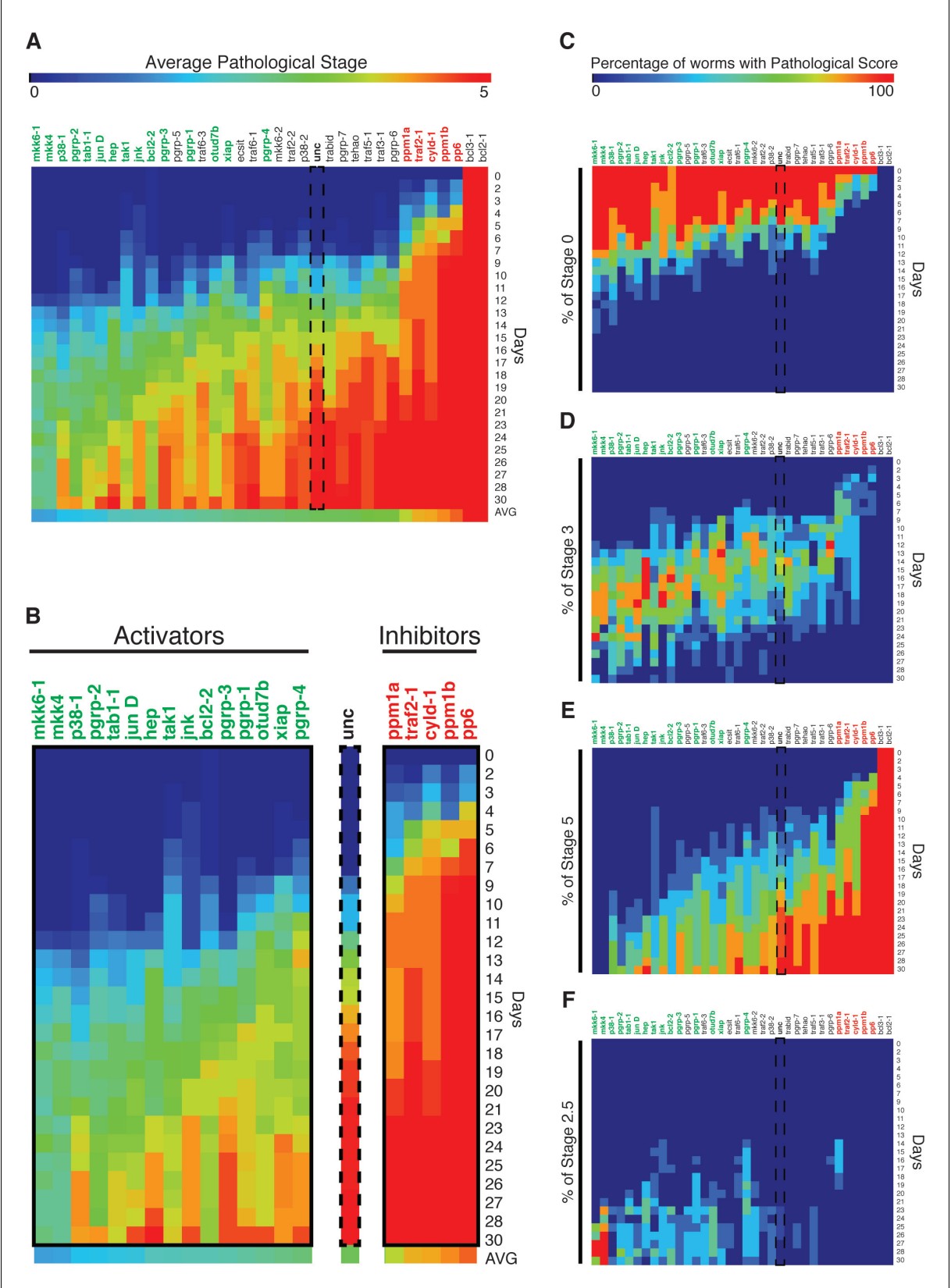

**Figure 5.** Heatmap depicting results of an RNAi screen for mediators of pathological progression following bacterial infection in planaria. (**A**) Heatmap depicting average pathological scores for each RNAi treatment following 2e8 CFU/ml *Pseudomonas* infection (n = 5–12 worms per condition). Days

*Figure 5 continued on next page*

*Figure 5 continued*

versus average pathological score over time are aligned along the y-, and x- axes, respectively. *Unc* control sample is indicated by a dashed box. RNAi-targeted genes that significantly reduce or enhance pathological progression are highlighted in green and red, respectively (2-way ANOVA p<0.05). (B) Focus on RNAi of genes that result in a significant reduction (activators) or enhancement (inhibitors) or pathological progression. (C–F) Heatmaps depicting the percentage of worms exhibiting pathological stage (C) 0, (D) 3, (E) 5, and (F) 2.5 for each RNAi treatment following *Pseudomonas* infection. Ordering and significance based on average pathological score over time are maintained for reference.

The following source data is available for figure 5:

**Source data 1.** Homologous innate immune and inflammatory genes from RNAi screen.
**Source data 2.** Pathological scores of worms in RNAi screen for mediators of infection induced tissue degeneration.

were analyzed at a tissue degeneration 'tipping point' at which point the following occurred: infected control worms developed anterior lesions (stage 2, Mild), RNAi of single activators resulted in posterior or no lesions (stage 0–1, None), and RNAi of single inhibitors produced head regression and/or lysis (stage 3–5, Severe) (*Figure 6B*). The resulting phenotypic stages of worms given each of 18 combinations of inhibitor and activator dsRNA were analyzed (*Figure 6C,D*, *Figure 6—source data 1*). Median pathological scores were used to order components in an effort to ascertain phenotypic outcomes while mitigating changes due to variation in phenotypic penetrance.

After 12 days of infection the median scores of worms given control dsRNA alone, activator dsRNA mixed with control dsRNA, or inhibitor dsRNA mixed with control dsRNA were 2, 0, and 3 or 4, respectively (*Figure 6C,D*). Introduction of activator dsRNA in combination with inhibitor dsRNA yielded clear and largely binary shifts in pathologic bacterial responses. Addition of either *mkk6-1, p38-1, tak1*, and *jnk* dsRNA rescued head regression and lysis observed in worms following RNAi of *pp6* alone. In contrast, knockdown of *pp6* in combination with *mkk4* phenocopied *pp6* dsRNA treatment while knockdown with *jun D* yielded a partial rescue. RNAi of *ppm1b* in combination with *mkk4, tak1*, and *jnk* phenocopied the bacterial-induced head regression observed in worms with *ppm1b* RNAi alone. In contrast, RNAi of *ppm1b* in combination with *mkk6-1, p38-1*, and *jun D* rescued head regression and phenocopied activator dsRNA treatment alone. Finally, head regression induced by *cyld-1* knockdown alone could be largely rescued by simultaneous knockdown of *mkk6-1* or *p38-1* but not *mkk4* or *tak1*. Curiously, combinatorial knockdown of *cyld-1* with *jun D* or *jnk* actually accelerated tissue degeneration suggesting additional roles for these components in other aspects of tissue homeostasis.

The results of our analyses allowed us to order these components with respect to their roles in mediating anterior tissue degeneration in response to infection: *mkk4, pp6, tak1, jnk, ppm1b, cyld-1, jun D, mkk6-1*, and *p38-1*. Remarkably, the Phenotype Hierarchy of these components is largely consistent with the TAK1 signaling pathway in humans (*Figure 6A,E*). Notably, there are some key distinctions. Given that we did not observe any effects of *mkk4* dsRNA when combined with any inhibitor dsRNA tested, our analysis placed *mkk4* either upstream of or lateral to these other components with respect to tissue degeneration. Additionally, our analysis placed *cyld-1* further downstream in the tissue degeneration process relative to its role in signaling in *H. sapiens*. The homologous deubiquitinating enzyme CYLD regulates ubiquitin mediated signaling and proteolysis. It is possible that its position within this Phenotype Hierarchy reflects a role in deubiquitination of proteins targeted for proteosomal degradation that are downstream of *jnk* rather than or in addition to targeting the homolog of the ubiquitin dependent kinase TAK1 (*Reiley et al., 2007*; *Xue et al., 2007*).

While the phenotype hierarchy derived from our combinatorial RNAi analysis is internally consistent and largely resembles the TAK1 signaling pathway in *H. sapiens*, one must take caution in extrapolating that this relationship represents the TAK1 signaling pathway in *S. mediterranea*. Traditionally, studies in more genetically amenable model organisms have utilized both epistatic analysis with null alleles to order genes and co-immunoprecipitation to demonstrate protein-protein interactions in the elucidation of signaling pathways. Thus, it remains to be determined to what extent the Phenotype Hierarchy uncovered here reflects the true *S. mediterranea* TAK1 signaling cascade.

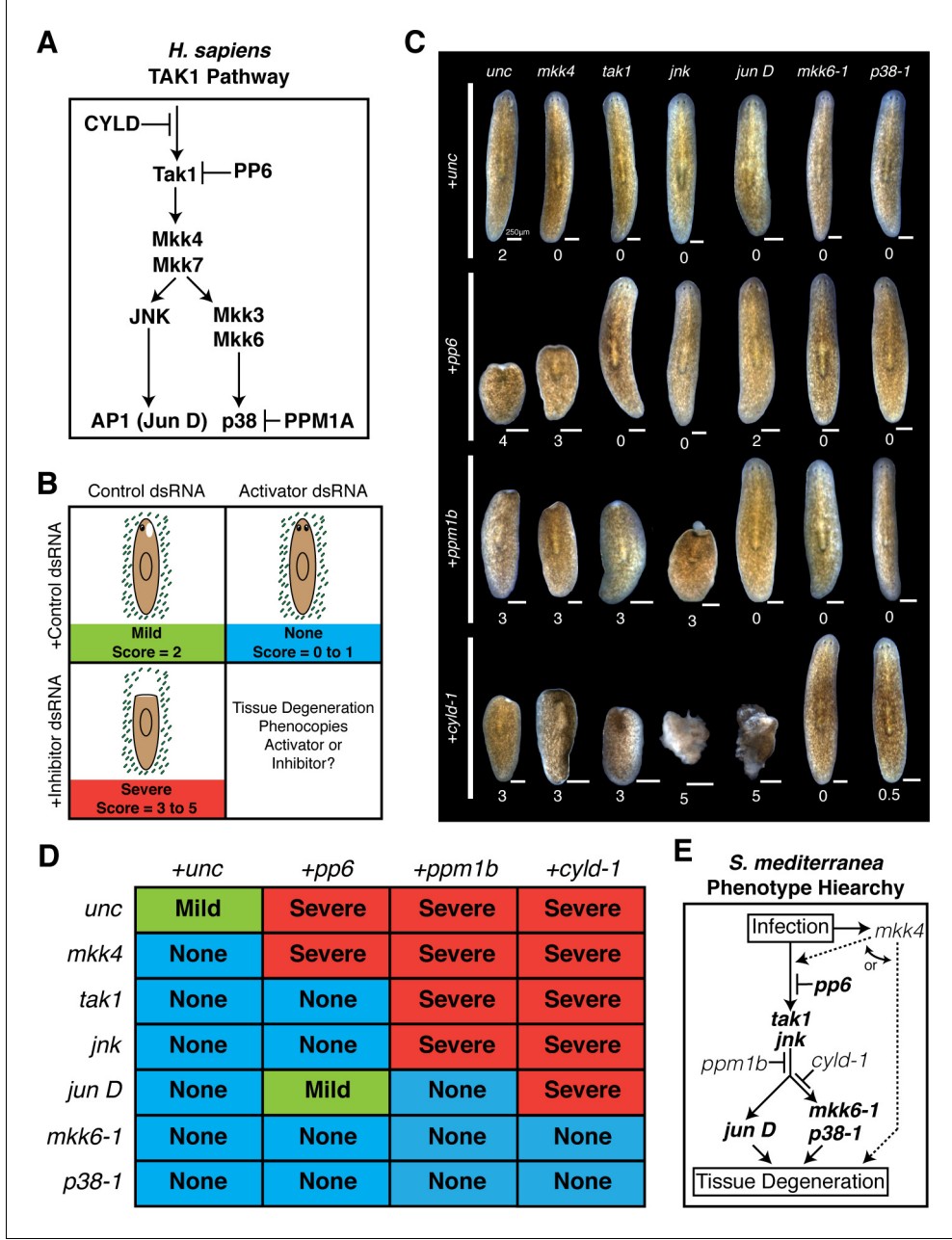

**Figure 6.** Epistatic analysis of activators and inhibitors of planarian pathological progression. (**A**) Diagram depicting relevant components of the TAK1 pathway in *Homo sapiens*. (**B**) Diagram depicting combinatorial RNAi experiment and assay of tissue degeneration outcome. (**C**) Representative images and median pathological stage of worms following RNAi treatment with each combination of 6 activators and 3 inhibitors 12 days post *Pseudomonas* infection (n = 6–24). (**D**) Table of phenotypic outcomes following combinatorial RNAi treatment and *Pseudomonas* infection. (**E**) Diagram depicting phenotypic hierarchy of the mediators of pathological progression in *Schmidtea mediterranea* (bold = order is consistent with *Homo sapiens* TAK1 Pathway).

The following source data is available for figure 6:

**Source data 1.** Pathological scores of worms during combinatorial RNAi analysis.

## Distinct roles of TAK1/MKK/p38 signaling components during regeneration in the absence or presence of infection

Following the identification of conserved TAK1/MKK/p38 signaling components in the mediation of tissue degeneration planaria in response to infection, we explored the roles of these components in amputation-induced regeneration. As previously described, infection with *Pseudomonas* compromises phenotypic regeneration of lost tissues and remodeling of existing tissues to various extents and in a dose dependent manner (*Figure 2F*). We examined the effects of RNAi knockdown of *mkk6-1, mkk4, p38-1, jun D, tak-1, jnk*, and *pp6* on regeneration in the presence or absence of infection (*Figure 7*). The concentration of dsRNA for *mkk6, mkk4, p38-1, jun D, tak-1, jnk* was doubled to increase the phenotypic penetrance while the concentration of dsRNA of *pp6* was halved to ensure the absence of any tissue degeneration during the initial period of infection prior to amputation. RNAi treated worms were infected with *Pseudomonas* for 2 days and then amputated above and below the pharynx to generate head, trunk, and tail fragments. Resulting phenotypic tissue regeneration was recorded 2 weeks later in comparison to uninfected worms for control and RNAi treated worms (*Figure 7A*). For quantitative characterization of RNAi effects, we categorized the resulting observed phenotypes and reported the proportion of regenerating fragments displaying the following: normal regeneration, abnormal regeneration, lesions, tissue regression, and lysis (*Figure 7B*).

The vast majority of head, trunk, and tail fragments regenerated normally following RNAi of target genes in the absence of infection (*Figure 7A,B*). Notably 20% of trunk and tail fragments lysed after *pp6* RNAi and 40–60% of trunk fragments exhibited abnormal regeneration (forked posterior balstemas) following *mkk4* and *jnk* RNAi. Knockdown of other positive regulators within this signaling module (*mkk6-1, p38-1, jun D,* and *tak1*) did not phenocopy these regeneration defects so they are likely the result of an additional role for *jnk*. This is consistent with previous studies highlighting defects in planarian regeneration following *jnk* RNAi (*Almuedo-Castillo et al., 2014*; *Tejada-Romero et al., 2015*). The relatively minor defects we observed may reflect different extents of *jnk* knockdown as the result of direct injection versus feeding of dsRNA.

In contrast to the relative lack regeneration phenotypes observed in the absence of infection, RNAi effects were much more pronounced in regenerating fragments from infected worms. Overall, RNAi of the activators largely prevented infection-associated defects in regeneration while RNAi of the inhibitor *pp6* exacerbated them (*Figure 7A,B*). Following infection, 100% of regenerating head fragments in control worms lysed while RNAi of *mkk6-1, p38-1,* and *jnk* not only prevented tissue lysis but re-established normal regeneration for 80% of the fragments. Similarly, RNAi of *mkk4, tak1,* and *jun D* reduced infection-associated defects in head fragment regeneration observed in controls. RNAi of *p38-1* or *tak-1* prevented infection-associated regenerative defects in tail fragments while *mkk6-1* or *jun D* RNAi reduced them. Regenerating trunk fragments were relatively resistant to the effects of the infection and RNAi of activators could largely reverse the minor defects observed in controls. Consistent with its role as an inhibitor of TAK1/MKK/p38 signaling, RNAi of *pp6* exacerbated infection-associated defects resulting in lysis of 100%, 60%, and 100% of regenerating head, trunk, and tail fragments, respectively (*Figure 7A,B*). These results demonstrate that conserved TAK1/MKK/p38 signaling components also play a pivotal role in determining regenerative outcomes induced by amputation during infection in a manner consistent with the positional sensitivity of these regenerating fragments along the A/P axis.

## Identification of an antibody labeling in situ Phospho-p38 signaling in planaria

WISH analysis of members of the TAK1/MKK/p38 signaling pathway revealed that these components were expressed in largely overlapping patterns of tissues along the planarian body axis. The most prominent expression pattern was evident in the planarian gut which was shared in both ordered activators (*tak1, jnk, jun D, mkk6-1,* and *p38-1*) and inhibitors (*pp6, ppm1b,* and *cyld-1*) (*Figure 8A*, *Figure 8—figure supplement 1A,B*). Weaker staining was also visible in the mesenchymal space between the gut and the epidermis with a slight anterior enrichment (*Figure 8A*, *Figure 8—figure supplement 1C*). While mRNA analysis is useful in identifying structures competent to respond to stimulation, it fails to resolve those cells or tissues dynamically responsive to infection. Fortunately, the phosphorylation-dependent signal transduction of the TAK1/MKK/p38 pathway presented an opportunity to analyze dynamic activation via phospho-antibody based detection of the downstream

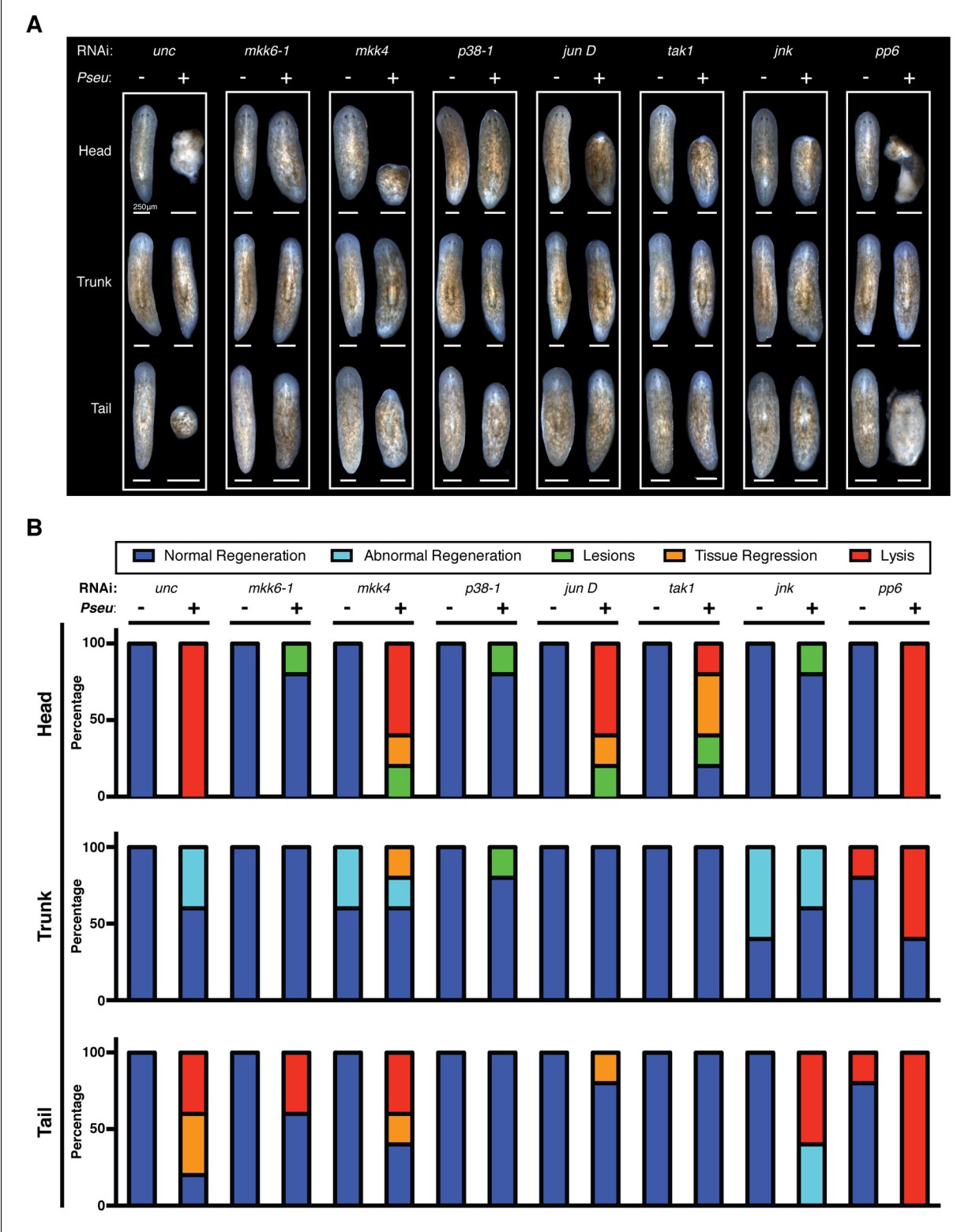

**Figure 7.** Effects of planarian pathological progression mediators on regeneration in the presence and absence of infection. (**A**) Representative images of regenerating head, trunk, and tail fragments following specified RNAi treatment in the presence or absence of *Pseudomonas* infection. Worms were amputated above and below the pharynx 2 days post infection and imaged 14 days post amputation (n = 5, non-lysed fragments shown when present). (**B**) Quantitation of phenotypes of regenerating head, trunk, and tail fragments following specified RNAi treatment in the presence or absence of

Figure 7 continued

*Pseudomonas* infection (dark blue = normal regeneration, light blue = abnormal regeneration, green = lesions, orange = tissue regression, red = lysis). Animals were scored at the same time when representative images were taken at 16dpi 14dpa.

target, p38. The human homolog of the kinase p38 is activated via phosphorylation at Thr180 and Tyr182, a region highly conserved in planaria (*Figure 8—figure supplement 2A*) (*Han et al., 1994*). We tested one polyclonal and one monoclonal phospho-p38 (Thr180/Tyr182) antibody for reactivity against the planarian phospho-epitope. UV treatment is a common inducer of P-p38 and western blot analysis revealed both tested antibodies reacted to an induced band at 45kDa corresponding to the size of planarian p38 (*Figure 8—figure supplement 2B*). In situ antibody staining revealed that both polyclonal and monoclonal antibodies recognized discrete cells above the pharynx (*Figure 8B*, *Figure 8—figure supplement 2C*). Initially tested bleaching treatments that were conducive to WISH protocols yielded high background fluorescence in the epidermis precluding analysis of reactivity in this peripheral tissue at this point (*Figure 8—figure supplement 2C*). Treatment with UV irradiation induced a broad antibody staining pattern throughout the gut, mirroring the localized expression of the *p38-1* transcript (*Figure 8A*, *Figure 8—figure supplement 1A*). We utilized the monoclonal antibody for subsequent analyses since it produced stronger staining.

We tested antibody specificity by determining if RNAi against planarian *p38-1* could eliminate the observed signal (*Figure 8—figure supplement 2D*). Worms fed beef liver pasted mixed with in vitro synthesized dsRNA had a relative lack of discrete cell staining near the pharyngeal intestinal border, while worms fed *E.coli* producing dsRNA had robust staining in the anterior gut. We found that in addition to UV irradiation, amputation of worms produced an induction of signal in discrete cells throughout the gut within 30 min. Under all observed instances of signal induction (*E. coli* feeding, amputation, and UV irradiation) we found no detectable staining after *p38-1* RNAi (*Figure 8—figure supplement 2D*). Thus, we conclude that this antibody specifically recognizes P-p38 in planaria.

## P-p38 signaling is dynamically modulated in the gut in response to infection or injury

Using the characterized P-p38 antibody, we analyzed phospho-signaling in the planarian gut during infection. Prior to *Pseudomonas* infection, we observed both P-p38$^+$ cells that co-labeled with gut marker *mat1* as well as P-p38$^+$*mat1*$^-$ cells adjacent to the gut (*Figure 8B,C*). P-p38$^+$ cells were relatively constant for the first 3 days of infection (*Figure 8C*). Six days after infection, large nuclear dense mounds consisting of P-p38$^+$ cells could be observed in the anterior lumen of the gut with more intense lumenal staining also seen in the posterior and regions (*Figure 8C*, *Figure 8—figure supplement 2E*). Eight days after infection the gut appeared more distorted and P-p38 staining had consolidated to distinct cells throughout the gut (*Figure 8C*, *Figure 8—figure supplement 2E*).

We compared the kinetics of P-p38 activation in response to injury versus bacterial infection. We observed a robust P-p38 staining throughout the interior gut of the head, trunk, and tail fragments within 5 min of amputation overlapping with lumenal nuclear dense mounds (*Figure 8D*, *Figure 8—figure supplement 2F*, *Figure 8—figure supplement 3*). Thirty five to sixty minutes after amputation, signaling coalesced into discrete P-p38$^+$ cells throughout the gut of the resulting fragment (*Figure 8D*). Double labeling analysis revealed that these P-p38$^+$ cells co-expressed the gut markers *mat1* (low), *porc*, *hnf4*, and *nkx-2.2* (*Gurley et al., 2008*; *Wagner et al., 2011*; *Tu et al., 2015*) (*Figure 8E*). Gut injuries resulting from either lateral incision or needle poke both elicited a similar but more localized P-p38 activation within 30 min (Note: epidermal staining in lateral wound could not be distinguished from background) (*Figure 8—figure supplement 2G*). The amputation induced P-p38 staining pattern persisted for 24 hr and by one week regenerating worms largely re-established the discrete P-p38 staining pattern observed in intact worms (*Figure 8D*). Quantitative image analysis confirmed that while infection resulted in an increase in P-p38 fraction volume over time, amputation induced a sharp increase in both P-p38 fraction volume and average intensity followed by a gradual decrease (*Figure 8F,G*).

At present, it is difficult to distinguish the extent to which P-p38 signaling in the gut is directly activated as a result infection versus indirectly activated by damage resulting from infection. Conversely, it is similarly difficult to evaluate the extent to which tissue injury that results in the invasion

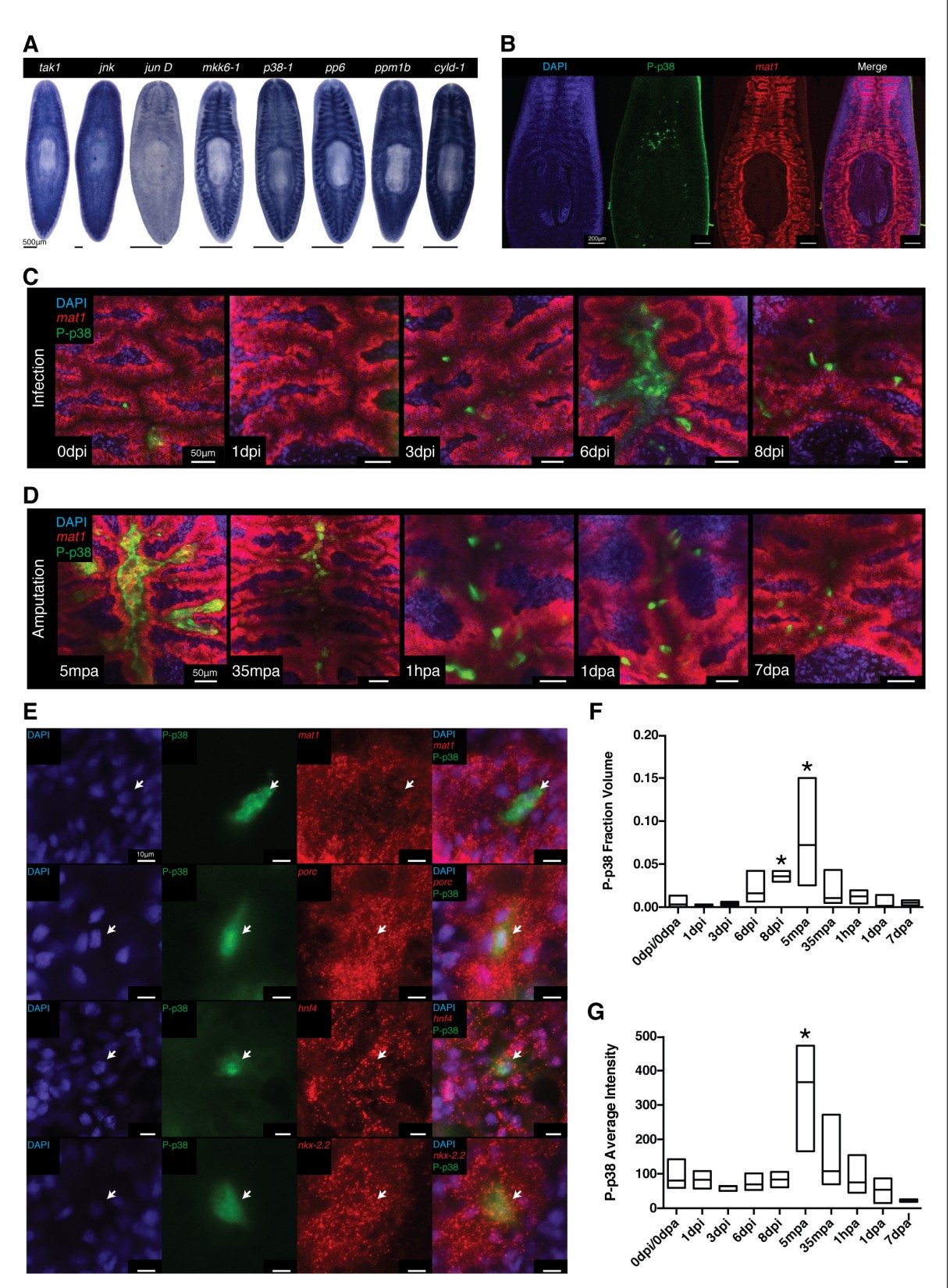

**Figure 8.** Signaling dynamics in gut tissue during infection and regeneration visualized with a phospho-p38 antibody. (**A**) Colorimetric WISH of genes comprising the TAK/MKK/p38 signaling module (n > 3 worms). Combinatorial fluorescent whole mount ISH and immuno-labeling of gut marker *mat1*
*Figure 8 continued on next page*

*Figure 8 continued*

and phospho-p38 (P-p38) (B) prior to stimulation, (C) following infection with 2e8 CFU/ml Pseudomonas, or (D) after amputation (n = 2–11 worms). (E) Higher magnification images of P-p38$^+$ cells co-labeled with gut markers *mat1, porc, hnf4*, and *nkx-2.2* following amputation (35-65 mpa). Quantification of P-p38 (F) Fraction Volume and (G) Average Intensity following infection or amputation (* = t-test p<0.05).

The following figure supplements are available for figure 8:

**Figure supplement 1.** Co-expression of TAK1/MKK/p38 pathway components.

**Figure supplement 2.** Validation of an antibody that recognizes planarian P-p38.

**Figure supplement 3.** High resolution analysis of P-p38 signaling in gut tissue following amputation.

of bacterial populations normally excluded from internal tissue structures elicits activation of P-p38. In either case, our data indicate that the downstream kinase of the TAK1/MKK/p38 signaling module is dynamically modulated in response to infection and/or injury within the planarian gut

## P-p38 signaling selectively initiates apoptosis in response to infection

Phospho-signaling analyses revealed that p38 was activated within the planarian gut in response to infection or injuries. Curiously, RNAi of the inhibitors *pp6* and *cyld-1* did not elicit intense P-p38 staining in the gut of intact worms prior to stimulation or during the first 3 days of infection even though the appearance of lesions and head regression immediately followed (data not shown). As previously indicated, initial P-p38 staining experiments utilizing bleaching methods conducive to WISH elicited background staining in peripheral tissues (*Figure 8—figure supplement 2C*). Use of an alternative bleaching solution eliminated background, revealing a previously unresolvable anterior/dorsal enriched mesenchymal/epidermal P-p38 staining pattern that was induced by *Pseudomonas* infection (*Figure 9A*, *Figure 9—figure supplement 1A,B*). P-p38$^+$ cells were relatively absent prior to infection but could be visualized in anterior regions within 24 hr of infection and further increased in frequency over the next two days. A lower frequency of P-p38$^+$ cells was observed in the posterior region with little to no signal in the mid body region. This signal represents the earliest observed P-p38 activation in response to infection in a pattern that overlaps with and precedes the occurrence of anterior lesions and eventual head regression (*Figure 2B*). Furthermore, its localization correlates with the differential sensitivity of regenerating tissue fragments to *Pseudomonas* infection (*Figure 2F*). We analyzed levels of apoptosis in response to *Pseudomonas* infection and observed a similar anteriorly biased induction of apoptosis that coincided with P-p38 activation (*Figure 9B*).

We determined the role of TAK1/MKK/p38 signaling in the observed infection induced apoptosis with a focus on the anterior region of the worm. We analyzed the results of ectopic induction of this pathway via RNAi of the negative regulators *pp6* and *cyld-1*. RNAi of either of these genes resulted in elevated numbers of anteriorly biased P-p38$^+$ and TUNEL$^+$ cells both prior to and following *Pseudomonas* infection (*Figure 9C–F*). Additionally, *cyld-1* RNAi resulted in pre-emptive increase in cell proliferation that mirrored the levels observed in control infected worms (*Figure 9—figure supplement 1C,D*). These data support a role for TAK1/MKK/p38 signaling in the control of both a general proliferative response and an apoptotic response during infection. The frequency of P-p38$^+$ cells was consistently lower than that of TUNEL$^+$ cells with only 1–8% overlap depending on RNAi treatment or days post infection (*Figure 9E,F*, *Figure 9—figure supplement 1E*, *Table 3*). Interestingly, we observed a higher frequency of TUNEL$^+$ cells that were proximal to P-p38$^+$ cells (ranging from 12–80%), suggesting a possible extrinsic, rather than an intrinsic role for the mediation of apoptosis (*Figure 9—figure supplement 1F*, *Table 3*).

We next determined whether TAK1/MKK/p38 signaling effects on apoptosis and proliferation were general or specific to infection. Previous research has demonstrated that tissue amputation elicits a robust induction of both proliferation and apoptosis (*Pellettieri et al., 2010*; *Wenemoser and Reddien, 2010*). We analyzed the effects of RNAi of both activators and inhibitors of the TAK1/MKK/p38 pathway on amputated tail fragments in the absence of infection. We observed no effects of RNAi of any TAK1/MKK/p38 signaling component on the proliferative burst following amputation (*Figure 9—figure supplement 1G*). To our surprise, amputation induced

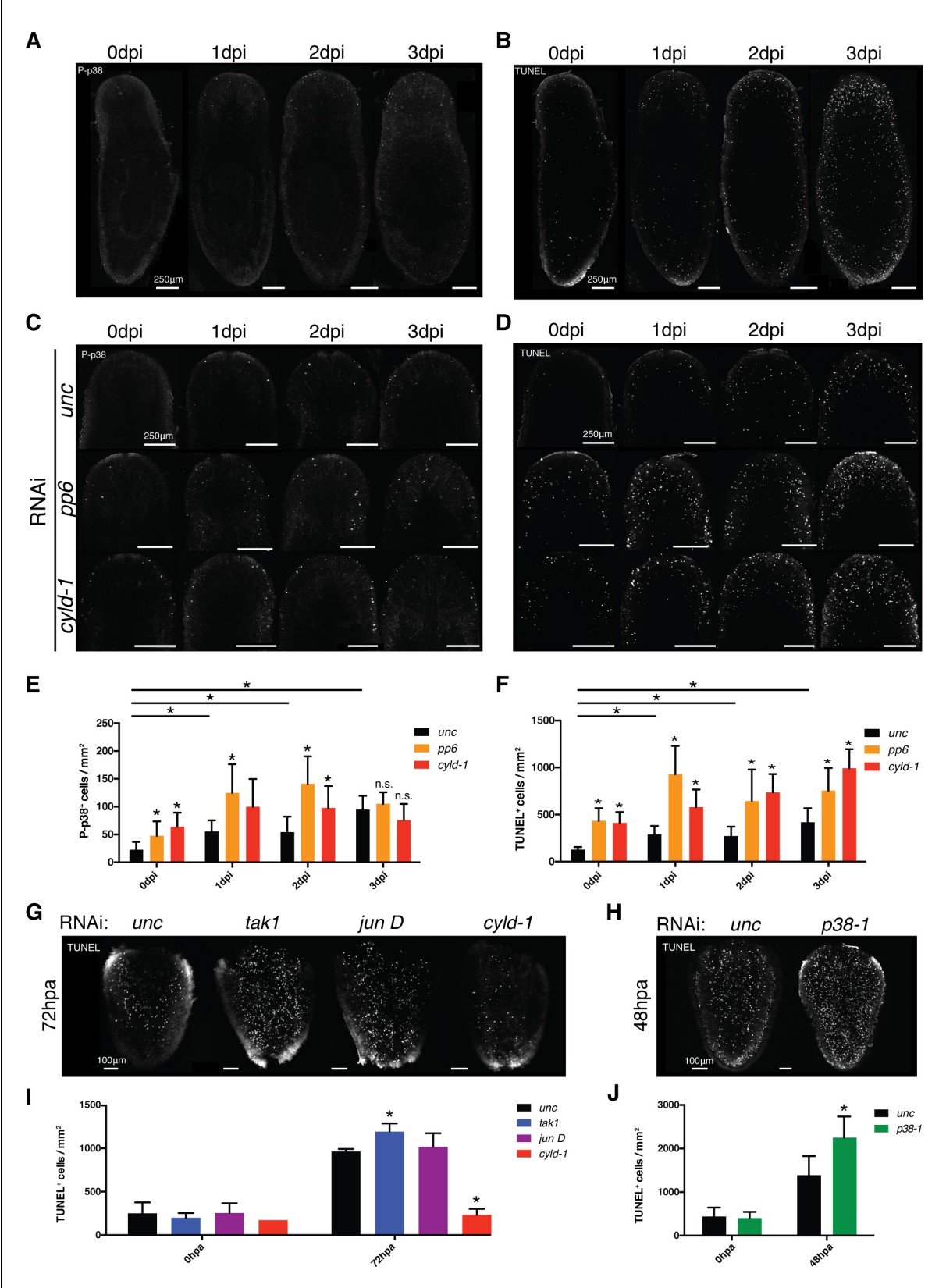

**Figure 9.** TAK1/MKK/p38 signaling mediates contrasting regulation of apoptosis in infection versus regeneration. (**A**) P-p38 staining or (**B**) TUNEL of *Unc* RNAi worms following *Pseudomonas* infection. Focus on the effects of *pp6* and *cyld-1* RNAi on (**C**) P-p38 signaling and (**D**) TUNEL in the anterior *Figure 9 continued on next page*

*Figure 9 continued*

regions of planaria during *Pseudomonas* infection. Quantification of RNAi effects on (E) P-p38[+] and (F) TUNEL[+] cells in the anterior during infection (n = 5–9). Confocal images of TUNEL in amputated tail fragments following RNAi of TAK1/MKK/p38 signaling components. RNAi effects assayed at (G) 72 hr or (H) 48 hr post-amputation (two independent experiments). (I, J) Quantification of RNAi effects on corresponding TUNEL experiment (n = 1–7) (* = t-test p<0.05).

The following figure supplement is available for figure 9:

**Figure supplement 1.** Effects of TAK1/MKK/p38 signaling on proliferation and apoptosis.

apoptosis was increased by RNAi of *tak1* or *p38-1,* and decreased by RNAi of *cyld-1* (*Figure 9G–J*). This stands in direct contrast to the observed effects of ectopic activation of this pathway in enhancing infection-induced apoptosis (*Figure 9D,F*). Interestingly, *tak1* and *p38-1* rnai worms all exhibited phenotypically normal regeneration in the absence of infection (*Figure 7*), suggesting that the observed enhancement in apoptosis is relatively innocuous during normal regeneration. Overall, our data indicate that TAK1/MKK/p38 signaling has dual contrasting roles in the control of apoptosis during either infection or regeneration.

## Downstream TAK1 signaling components have a role in the planarian immune response

We evaluated the potential role of TAK1 signaling pathway homologs in the planarian immune response. Planaria were infected and maintained in planaria water containing 2e8 CFU/ml *Pseudomonas* without reinfection to assay clearance of the initial bacterial dosage. Subsequent changes in bacterial levels and tissue degeneration of control and activator/inhibitor RNAi worms were monitored over time. *Pseudomonas* levels in control RNAi worms increased to ~1e5 CFU/worm by 1 to 3 days and 2e5 CFU/worm by 6 days (*Figure 10A–D*). During this time worms exhibited no phenotypic signs of tissue degeneration (*Figure 10E*). By 12 days post infection, nearly all control worms exhibited head regression or anterior lesions, coincident with a ~80% decrease in *Pseudomonas* levels per worm (*Figure 10A–E*). RNAi of the activator *jun D* significantly increased bacterial load over time while RNAi of *p38-1* had no overall significant effects on *Pseudomonas* levels (*Figure 10A, B*). Importantly, *jun D* RNAi worms still exhibited a reduction in bacterial levels from day 6 to day 12, suggesting that *jun D* independent immune responses also mediate bacterial clearance (*Abnave et al., 2014*). The roles of the inhibitors *pp6* and *cyld-1* in the immune response were less clear as RNAi of either of these components resulted in a non-significant overall increase in

**Table 3.** Quantitation of overlapping and proximal P-p38 and TUNEL signal.

| RNAi condition | Days post infection | % of P-p38 signal overlapping with TUNEL | % of P-p38 signal proximal to TUNEL |
|---|---|---|---|
| *unc* | 0 | 1.27 | 12.66 |
| *cyld-1* | 0 | 5.78 | 45.66 |
| *pp6* | 0 | 4.29 | 67.86 |
| *unc* | 1 | 1.16 | 39.31 |
| *cyld-1* | 1 | 4.52 | 62.05 |
| *pp6* | 1 | 8.87 | 79.31 |
| *unc* | 2 | 1.90 | 34.29 |
| *cyld-1* | 2 | 3.48 | 60.65 |
| *pp6* | 2 | 5.73 | 57.96 |
| *unc* | 3 | 2.55 | 44.19 |
| *cyld-1* | 3 | 3.98 | 54.23 |
| *pp6* | 3 | 8.29 | 57.46 |

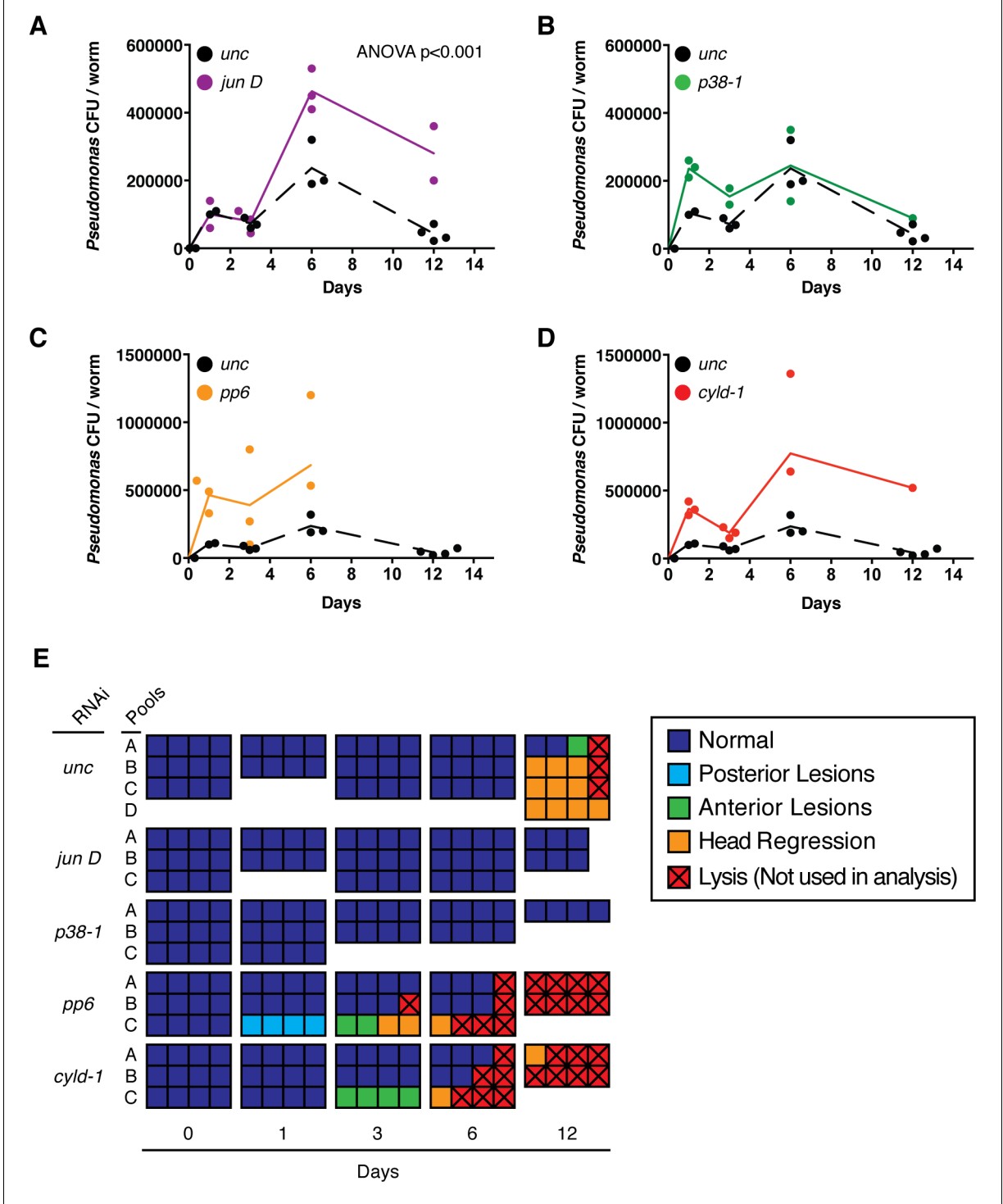

**Figure 10.** Effects of TAK1 components on the immune response. Bacterial CFU/worm following infection with a single dose of 2e8 CFU/ml *Pseudomonas* and RNAi of (**A**) *jun D*, (**B**) *p38-1*, (**C**) *pp6*, or (**D**) *cyld-1* (n = 1–4 pools of 1–4 worms each). (**E**) Pathological state of individual infected worms (squares) pooled for bacterial CFU/worm analysis.

*Pseudomonas* over time (*Figure 10C,D*). We hypothesize that increased basal apoptosis and induced tissue degeneration in these RNAi conditions compromises barrier epithelia, leading to a larger influx of bacteria and complicating analysis of the immune response (*Figures 9*, *10E*).

These data suggest that the planarian immune response is normally coordinated with but not the result of induced apoptosis and tissue degeneration. While control RNAi worms exhibit the largest decrease in *Pseudomonas* levels following phenotypic tissue degeneration (*Figure 10A–E*), RNAi of *p38-1* results in an identical reduction of bacterial levels in the absence of phenotypic tissue degeneration (*Figure 10B,E*). Similarly, *pp6* and *cyld-1* RNAi enhanced tissue degeneration but not bacterial clearance (*Figure 10C–E*). Thus, *Pseudomonas* infection induces a *jun D* and *p38-1* dependent tissue degeneration response in conjunction with a *jun D* dependent, *p38-1* independent immune response.

## Discussion

In this study, we have utilized a combination of novel culture methods, bacterial 16 s rDNA sequencing, transcriptional profiling, RNAi screening, and in situ phospho-signaling visualization to illustrate how microbial dysbiosis results in the amplification of pathobionts, compromising tissue homeostasis and regeneration potential via the activation of TAK1/MKK/p38 signaling. Sequencing of the planarian microbiome revealed a high Bacteroidetes to Proteobacteria ratio in healthy animals. Perturbation such as tissue amputation or changes in culture conditions resulted in a reciprocal contraction of the Bacteroidetes and an expansion of the Proteobacteria populations. Coincident with these microbial shifts, intact worms displayed increased susceptibility to the development of visible tissue homeostatic defects. Isolation of emergent bacteria during this period yielded a strain of *Pseudomonas* which itself was sufficient to induce tissue degeneration in a morphologically stereotypical progression. Development of novel anti-septic culture methods permitted a candidate RNAi screen focusing on the role of conserved mediators of innate immunity and inflammation in mediating this response to infection. We identified TAK1/MKK/p38 signaling components underlying tissue degeneration and compromised regeneration capacity during infection. Interestingly, while TAK1/MKK/p38 signaling enhanced apoptosis during infection, it repressed apoptosis during tissue regeneration in the absence of infection. Additionally, these tissue degeneration responses were independent of an immune response mediated by *jun D*, an AP-1 family homolog downstream of TAK1. Taken together, this study demonstrates that Proteobacteria expansion elicits distinct activation of TAK1/MKK/p38 signaling which in turn acts as an impediment to complex tissue regeneration.

### An invertebrate microbiome with composition and dynamics analogous to the mammalian lower intestinal tract

This study elucidates the heretofore unknown *S. mediterranea* microbiome. We observed how microbial composition dynamically shifted in response to changes in culture conditions or during regeneration (*Figure 1*, *Figure 1—figure supplement 1*). The ratio of Bacteroidetes to Proteobacteria within the planaria from the recirculation culture system mirrors the distribution of these phyla within the human lower intestinal tract (*Human Microbiome Project Consortium, 2012*). This phyla composition is not observed in other invertebrate (*e.g.*, *C. elegans* and *D. melanogaster*) or even vertebrate (*e.g.*, *D. rerio*) model systems that are amenable to large scale genetic screens and have played a pivotal role in shaping our understanding of immunity and host-microbe interactions (*Apidianakis and Rahme, 2011*; *Roeselers et al., 2011*; *Novoa and Figueras, 2012*; *Lee and Brey, 2013*; *Wong et al., 2013*; *Ermolaeva and Schumacher, 2014*; *Berg et al., 2016*). An enrichment in Bacteroidetes in *S. mediterranea* is significant, as this phylum represents a diverse array of key symbionts known to support proper tissue homeostasis, the development of the immune system, as well as the prevention of inflammatory diseases in mammals, and, intriguingly, have even been linked to the early evolution of multicellular organisms (*Rakoff-Nahoum et al., 2004*; *Mazmanian et al., 2005*, *2008*; *Alegado et al., 2012*). Furthermore, changes in culture conditions or tissue transection results in an expansion in Proteobacteria, a phenomenon linked to a myriad of human inflammatory disorders (*Frank et al., 2007*; *Shin et al., 2015*). Models studying this phenomenon in subsets of genetically-altered or aged individuals have proven instrumental in the analysis of the effects of this expansion, but have been less useful in dissecting factors mediating its causation (*Carvalho et al., 2012*; *Clark et al., 2015*). Our ability to induce shifts in the microbiota of wild type animals, uniquely positions planaria as a model for the study of the interrelationship between endogenous bacterial dynamics, tissue homeostasis, and complex tissue regeneration with implications for human health.

## Contribution of the microbiome to tissue homeostasis and regeneration and its implications for planarian research

Our study has highlighted the impact that the microbiota can have on tissue homeostasis and regeneration in planaria. Maintenance and care of planarians using traditional static methods can yield variable bacterial levels and composition over time, potentially influencing experimental outcomes. Furthermore, prolonged use of antibiotics can lead to the gradual accumulation of resistant pathogenic strains, such as *Pseudomonas*. Given that many published phenotypes describing gene knockdowns mirror some of the observed effects of bacterial infection (*Reddien et al., 2005*; *Forsthoefel et al., 2012*; *Labbé et al., 2012*; *Tu et al., 2012*), it will be important to examine the extent to which alterations in endogenous bacteria or stimulation of immune signaling underlie these phenotypes. This issue is particularly relevant to the common practice of feeding worms *E. coli* producing dsRNA as a means of gene function interrogation. As we have demonstrated, this method of RNAi elicits robust P-p38 activation (*Figure 8—figure supplement 2E*). To mitigate the issues of inconsistent bacterial composition and variable immune stimulation across experiments, we have developed an RNAi workflow using recirculation and flow culture systems (*Figure 4*, *Figure 4—figure supplement 1*). Previous screens examining genes involved in planarian regeneration have utilized either static cultured or *E. coli* fed worms of indeterminate bacterial content. It would be of interest to employ our novel culture systems to determine whether published phenotypes describing tissue homeostatic defects similar to those observed in this study have inadvertently uncovered genes involved in the interplay between the immune system and regeneration.

## TAK1/MKK/p38 signaling and the immune system

Our study revealed a distinct role for TAK1/MKK/p38 mediated cellular apoptosis during infection versus normal regeneration in planaria. Hyper activation of TAK1/MKK/p38 signaling via *pp6* or *cyld-1* RNAi elevated basal apoptotic rates and increased stem cell proliferation prior to infection (*Figure 9D,F*, *Figure 9—figure supplement 1C,D*). In contrast, TAK1/MKK/p38 signaling repressed apoptosis during regeneration in the absence of infection (*Figure 9G–J*). Importantly, *tak1*, *mkk6-1*, *p38-1*, and *jun D* RNAi worms exhibited phenotypically normal regeneration in the absence of infection, suggesting that this alteration in apoptosis did not ultimately hinder overall tissue morphogenesis and remodeling (*Figure 7*). It is unclear at this point whether these divergent outcomes of TAK1/MKK/p38 signaling are the result of unique co-stimulation or the activation of discrete cell types induced during infection versus amputation. Nevertheless, our study provides further evidence that TAK1/MKK/p38 signaling is multifaceted and can play supportive or antagonistic roles in tissue regeneration (*Zarubin and Han, 2005*; *Cuenda and Rousseau, 2007*; *Karin and Clevers, 2016*).

The role of TAK1/MKK/p38 in apoptosis during infection is not unique to planaria. In *C. elegans*, oral *Pseudomonas* infection initiates programmed cell death of gonadal cells in a p38-dependent manner (*Aballay and Ausubel, 2001*). In *Drosophila*, oral infection induces p38-mediated production of reactive oxygen species (ROS), damaging gut epithelia, and triggering intestinal stem cell activation (*Buchon et al., 2009*; *Ha et al., 2009*). Interestingly, attenuation of TAK1/MKK/p38 signaling in nematodes and flies largely compromises host survival to infection, but substantially enhances host survival in planaria (*Figure 5*) (*Kim et al., 2002*; *Buchon et al., 2009*; *Chen et al., 2010*). One explanation is that the persistence of a population of adult pluripotent stem cells capable of regenerating all missing tissues uniquely endows planaria with a resilience to somatic tissue damage. Planaria may utilize TAK1 innate immune signaling for a coordinate mounting of antibacterial responses while actively clearing infected tissue via apoptosis prior to the initiation of regeneration. Under our sustained infection protocol, pathogen levels were constantly maintained, resulting in failure of complete bacterial clearance and continuous TAK1/MKK/p38 mediated apoptosis. Once this level of tissue turnover outpaces the ability of neoblasts to replace these cells, tissue homeostasis and regeneration capabilities are compromised. Under this context attenuation of infection-induced TAK1/MKK/p38 signaling rescued tissue degeneration and restored regenerative potential.

## A case of tissue regeneration during chronic infection

The management and care of chronic non-healing wounds poses a substantial and sharply rising burden to our healthcare system and economy (*Sen et al., 2009*). Of interest is the observation that planaria are capable of resolving tissue degenerative wounds induced by sustained *Pseudomonas*

infection and regenerating lost anterior structures (*Figure 2C*, *Figure 2—figure supplement 1D*). Antibiotic resistant *Pseudomonas* is increasing in prevalence in chronic wounds and poses an impediment to effective healing (*Dowd et al., 2008*; *Fazli et al., 2009*; *Price et al., 2009*; *Goldufsky et al., 2015*). Further research is needed for the development of treatments by which to effectively manage infection and tissue repair. Limited regenerative capabilities combined with the severe lethality of *Pseudomonas* infection has precluded the study of this phenomenon in other model systems (*Kim et al., 2002*; *Vodovar et al., 2005*). In contrast, results from our RNAi screen have already indicated that *mkk4* and *mkk6-1* may play a role in inhibiting this post-infection regeneration in a *p38-1* independent manner (*Figure 5F*). The molecular mechanism by which this is accomplished is still unclear. Furthermore, the observation that this phenomenon was unique to *Pseudomonas* infection relative to the other pathogenic strains tested is equally fascinating. In plants, some species of *Pseudomonas* are capable of attenuating host immune response via secretion of a tyrosine phosphatase that inhibits signaling (*Macho et al., 2014*). It would be of great interest to compare the differential host responses of planaria to other pathogens or mutant *Pseudomonas* strains to determine unique gene regulatory networks supportive of the observed tissue regeneration during chronic infection.

## Conclusion

The novel tools and findings of this study not only open up new areas of inquiry for planaria as a model system, but also broaden our understanding of the relationship between endogenous microbiota, the innate immune system, and regeneration. We took advantage of the amenability of planarians to RNA-mediated genetic interference to identify mediators of *Pseudomonas*-induced tissue degeneration and its lytic versus regenerative outcomes. We identified a TAK1 innate immunity signaling module underlying tissue degeneration and compromising regeneration capacity during infection. Interestingly, while TAK1/MKK/p38 signaling enhanced apoptosis during infection, it repressed apoptosis during tissue regeneration in the absence of infection. Our data indicate that activation of this pathway has discrete and seemingly opposite roles in host immunity versus normal regeneration. Given the complex role of inflammation in either the hindrance or support of reparative wound healing and regeneration, *S. mediterranea* provides a basis for dissecting the duality of this evolutionarily conserved inflammatory signaling module in complex, multi-organ adult tissue regeneration. Furthermore, the conservation of bacterial phyla composition and dynamics we uncovered in this invertebrate make it the first non-murine model organism for the study of pathological shifts in endogenous bacteria with relevance to human disease. In summation, our work not only advances planarians as a unique and tractable in vivo model system for the molecular dissection of regeneration, but also opens the door to the identification and characterization of conditions that promote or inhibit the natural execution of regenerative processes.

## Materials and methods

### Planarian care in static or recirculation culture

*Schmidtea mediterranea* clonal CIW4 strain was maintained in 1X Montjuic salt as previously described for traditional static culture methods (*Newmark and Sánchez Alvarado, 2000*), supplemented with 50 µg/ml gentamycin where indicated.

A novel recirculation culture system was the predominant source of CIW4 planaria used in this study. The planarian recirculation culture system is composed of three culture trays (96' L × 24' W × 12' H) stacked vertically on top of each other over a sump. Planarian water flows from the sump pump through a chiller, canister filter, and a UV sterilizer into the top tray. Water subsequently flows down drains at the opposite ends of the source through the series of three trays. Water then re-enters the sump where it is filtered through two vertically stacked 400 µm and 200 µm sieves. Beyond the sieves, water is gravity fed and mechanically filtered through a set of filter/floss pads. Finally, water passes through Water Garden Oasis Pond Matrix and Kaldness media to remove nitrogenous waste. Water is then able to flow back through the chiller, canister filter, and UV sterilizer back into the top tray.

For the sexual S2F2 strain of planaria, worms were obtained from a fill and drain system. The system is composed of 32 tanks and a sump. All tanks drain into a common drain line, flowing into the

filter box resting on top of the sump. The sump contains a heavily aerated bed of Kaldness media for biofiltration. Water is next pumped through finer sets of sock filters (50 and 20 µm) and then through ultraviolet irradiation prior to being returned to the culture tanks. Each of the culture tanks utilizes a recurring fill and drain action to provide more frequent water turnover for each tank. Animals are housed within cylinders with porous mesh bottoms within these tanks.

## Planarian genomic DNA preparation for 16 s sequencing

For recirculation to static time course, 4 day starved worms were removed from the system and placed in containers filled with planaria water with or without the antibiotic gentamycin. For regeneration time course, 14 day starved worms were removed from recirculation culture at low density, washed after two days and transferred to petri dishes after 3 days. Twenty intact worms or 60 amputated tissue fragments were allocated per dish per time point. For intact worms, only worms that had not undergone fission were used for analysis. Worms were washed every 3–4 days after amputation. Wild worms were collected from sites in Sardinia in RNA later for sample storage.

DNA isolation was performed as follows. Negative controls from the initial DNA isolation were carried out for regeneration time course and wild worms. Collected worms were crushed in 100 µl in chilled lysis buffer containing 20 mM Tris pH 8.0, 100 mM NaCl, 50 mM EDTA, 2 mM spermidine, and 0.2 mg/ml Proteinase K, and 0.2% $\beta$-Me. Lysate was added to 800 µl of lysis buffer and warmed to 50C. 100 µl of 10% SDS was added and the tube was nutated at 50°C for 1 hr. 1 ml of phenol:chloroform:IAA was added to each tube and mixed on nutator for 10 min. Samples were spun at 4°C 1600 g for 5 min to get phase separation. The top layer was transferred to a new tube and 1 ml chloroform was added. Samples were spun again for phase separation and the top layer was transferred to a new tube. Samples were precipitated by adding 12% volume of 2.5 M NaAc and 1 volume isopropanol. After 1 hr incubation at −20°C, samples were pelleted by centrifugation and rinsed 3 times with 70% EtOH. The pellet was dried and resuspended in $H_2O$. Samples were RNAse treated in a solution of 5 mM Tris pH 8.0 and 1 µg/ml RNAse A for 15 min at room temperature. Samples were precipitated with 12.5% NaAc, 2.5 volumes of EtOH and incubation at −80°C for 1 hr. Samples were pelleted by centrifugation and washed 3 times with 70% EtOH. Pellets were dried and resuspended in $H_2O$.

## 16 s Metagenomic Sequencing

Sample preparation was performed as per manufacturer's instructions with some modifications. In brief, the V3 and V4 primer pair was used for template amplification using 2X KAPA HiFi Hotstart ReadyMix (KAPA Biosystems, Wilmington, MA) in PCR strip vials. Reactions were placed in a thermal cycler at 95°C 3 min, followed by 25 to 35 cycles of 95°C 30 s, 55°C 30 s, and 72°C 30 s with a 5 min 72°C extension. For static culture timecourse and regeneration timecourse, 35 and 32 cycles were used respectively. For wild Sardinia worm material, 25 cycles were initially used for all samples. Those that failed to amplify in addition to a control successfully amplified sample were rerun with 35 cycles. Negative controls of $H_2O$ were run in parallel from the start of library preparation for static time course sample set. Negative controls from the start of the DNA isolation were run for wild worm and regeneration sample sets. Following the initial PCR amplification, the template was purified using Agencourt AMPure XP beads and a magnetic stand. Next, indexing pcr was performed using the Nextera XT index kit (Illumina). Reactions were placed in a thermal cycler using the identical PCR program with only 8 cycles. A final round of cleanup with Agencourt AMPure XP beads was performed and libraries were pooled, requantified, and sequenced as 2 × 250 cycle paired reads on the Illumina MiSeq, using MiSeq Control Software v2.5.0.5. Following sequencing, the MiSeq Reporter v2.5.1.3 (recirculation to static time course experiment) or v2.6.2.3 (regeneration time course and individual wild worm experiments) Metagenomics workflow was used to de-multiplex reads for all libraries, generate FASTQ files, and perform taxonomic classification of the reads. Reads of negative controls were used to determine background bacterial amplification. After elimination of background levels, a low signal threshold was set at 10 reads per OTU.

For wild worms, 16 s rDNA was successfully amplified and sequenced for 4 out of 6 worms from Sardinia collection site#1 (wild worm#1 jj04-9, wild worm#2 jj04-11, wild worm#3 jj04-11, wild worm#4 jj04-13) and 1 out of 1 worms from collection site#2 (wild worm#5 jj08-1). A 5th worm from collection site #1 was also amplified and sequenced (jj04-10), but since this sample did not have a

negative control run in tandem during DNA isolation (as was performed for the other wild samples), it was not possible to determine potential contamination of the sample with bacteria from the lab. Therefore, this sample was not used for further analysis.

## Data visualization

Graphs were generated using Prizm software. Heatmaps were made using TM4 MeV. Venn Diagrams were generated using Venneuler.

## Microscopy

Images of colorimetric WISH samples, histology samples, confocal images, and live worm images were acquired using Zeiss Lumar, Leica DM 600B, LSM510-VIS, and Leica M205 microscopes, respectively. Individual tiled whole worm or 10X zoomed head images were acquired on a Perkin Elmer Ultraview spinning disk microscope for P-p38, TUNEL, and phospho H3 quantification. Stitching was performed using Fiji plugins combined with customized batch processing macros or wrapper plugins where necessary. Worms were segmented by DAPI labeling with custom plugins and spots were counted using the 'Find Maxima' function via batch processing macros. All macros and plugins are available at https://github.com/jouyun. For P-p38 average intensity and fraction whole quantification, spots were segmented and manually filtered from false worm boundary objects and used to measure spot area and compute intensities. The whole worm or amputated fragment was segmented using DAPI to compute the total volume.

## Bacterial colony forming unit (CFU) assay

The specified number of worms or fragments were washed three times and then crushed in 100 µl $H_2O$. Homogenate was serially diluted 4X by 10-fold and 25 µl was plated onto each quadrant of an LB plate. Plates were incubated at room temperature for 3–4 days and the morphology and the number of colonies was recorded. For identification of abundant strains, colony morphology was categorized by color, shape and size. The 16 s variable region from representative colonies of each category was PCR amplified with the following primers: forward primer 63f (5′-CAG GCC TAA CAC ATG CAA GTC-3′) and reverse primer 1378r (5′-GGG CGG WGT GTA CAA GGC-3′) (*Marchesi et al., 1998*). The resulting template was sequenced and nucleotide BLAST was performed for putative identification.

## Determining bacterial concentrations of isolated strains

Strains isolated from LB plating of planaria homogenate were identified by 16 s sequence homology and stored in glycerol stocks. Concentration was determined by diluting bacteria to a 600 nm absorbance of 1 and plating serial dilutions on LB plates. Average colony count across replicates was used to determine an absorbance to bacterial CFU/ml conversion factor.

## Bacterial infection experiments

Worms were transferred to petri dishes and allowed to rest overnight. Bacterial strains stored in glycerol stocks were used to inoculate LB for overnight culture at 30°C. Bacterial cultures were spun down, washed, and resuspended in planaria water at the specified concentrations using 600 nM absorbance and previously calculated bacterial CFU/ml conversion factors. Planaria were washed several times and bacteria resuspended in planaria water was added. Infection was refreshed every 3 to 4 days (unless specified otherwise) by washing worms several times, transferring to a new petri dish, and adding fresh bacteria resuspended in planaria water.

## Pathological scoring

Worms were ocularly scored every 1 to 3 days using a Zeiss Stemi 200-C stereo microscope using the following criteria. Worms exhibiting no phenotypic lesions were scored 0. Worms exhibiting lesions in either the mid-body or posterior were scored 1. Worms with lesions in the anterior but displaying at least one remaining photoreceptor were scored 2. Worms with anterior tissue degeneration that resulted in no remaining photoreceptors were scored 3. Worms with regressed anterior regions that displayed partially disrupted or lysed tissue were scored 4. Worms that fully lysed were scored 5. Worms that regressed heads and formed blastemas with both visible photoreceptors were

scored 2.5. Worms inadvertently lost or damaged during reinfection procedures and those that crawled out of sample dishes were not scored further.

## Histology preparation and staining

For histology preparation, animals were treated with 7.5% NAC (8 min), fixed in 4% PFA in PBSTx 0.3% (20 min), and then dehydrated by washing in 30%, 50%, and 70% methanol (5 min each). For paraffin embedding, animals were soaked in ethanol with 5% glycerol, washed in xylene (7 min) and clear-rite (2 × 7 min), and soaked in paraffin (2 × 14 min, then 2 × 30 min). After serial sectioning (6 μm thickness), slides were heated to 60°C for 20 min, deparaffinized with three 3 min washes in xylene, washed 3 × 1 min in 100% ethanol, then 80% ethanol, rinsed in tap water. Subsequent staining utilized the Leica Infinity system and was performed in a Leica Autostainer

For Alcian Blue/ Periodic acid-Schiff (AB/PAS) staining, hydrated slides were stained with 1% Alcian Blue made in 3% acetic acid for 15 min. Afterwards, slides were washed (2 min) in running tap water, rinsed in DiH$_2$O, incubated in 0.5% periodic acid (5 min), and rinsed in DiH$_2$O. Slides were then stained in Schiff's reagent (10 min) and rinsed under tap water (5 min). Nuclei were stained with hematoxylin (1 min) and rinsed in running tap water (2 min). Slides were placed in acid alcohol (1 min, Leica Infinity, Differentiator), rinsed in tap water (1 min), and incubated in Bluing Agent (1 min) followed by tap water rinse (1 min). Finally, the samples were dehydrated through washes in 80% ethanol, 4 × 100% ethanol, cleared in xylene, and mounted with coverslip onto slides.

For anti-*Pseudomonas* staining, antigen retrieval was performed using citrate buffer, pH 6.0 for 15 min at 95°C, cooled 20 min, and rinsed in DiH$_2$O. Slides were treated 30 min in Background buster (VWR #NB306) for 30 min and rinsed. Staining was performed with anti-*Pseudomonas fluorescens* antibody (abcam #25182) 1:750, overnight at 4°C, rinsed, stained 1 hr with donkey anti-goat 488 for 1 hr, and rinsed. Slides were mounted under coverslip with Fluoromount G with DAPI (VWR #102092–102).

## RNAseq

For RNAseq analysis, 4 biological replicates of 4 worms were collected at the specified times after the transition from recirculation to static culture in the presence or absence of 50 μg/ml gentamycin for TRIZOL isolation. mRNAseq libraries were generated from 500 ng of high quality total RNA, as assessed using the Bioanalyzer 2100 (Agilent). Libraries were made according to the manufacturer's directions for the TruSeq Stranded mRNA LT– set A and B (Illumina, San Diego, CA; Cat. No. RS-122-2101 and RS-122-2102) and using NEXTflex DNA Barcodes (BiooScientific, Austin, TX; Cat. No. 514104). Resulting short fragment libraries were checked for quality and quantity using a LabChip GX (Perkin Elmer) and Qubit Fluorometer (Life Technologies, Carlsbad, CA). Libraries were pooled, requantified and sequenced as 50 bp single reads on the Illumina HiSeq 2500 instrument using HiSeq Control Software v2.2.38. Following sequencing, Illumina Primary Analysis version RTA v1.18.61 and Secondary Analysis version CASAVA-1.8.2 were run to demultiplex reads for all libraries and generate FASTQ files. RNAseq analysis was performed as previously described using a minimum fold change of 1.4 and an adjusted p-value < 0.05 (*Tu et al., 2015*).

## Quantitative PCR

Reverse transcriptase (for *pgrp* genes) and quantitative PCR (for *pgrp* genes and 16 s rDNA) was performed using Superscript III (Invitrogen) and Fast SYBR Green Master mix (ThermoFisher) for the following gene targets: universal 16 s (f: 5'-GTG STG CAY GGY TGT CGT CA-3', r: 5'-ACG TCR TCC MCA CCT TCC TC-3'), *pgrp-1* (f: 5'-CTG CCA TCC GAT AAG ATG AGT T-3', r: 5'-TAT CGT TTC TCG TCG GCA TTT A-3'), *pgrp-4* (f: 5'-GAC TCT CGA TCC GAA AGT AGG A-3', r: 5'-GGG TTG TCC ATT CCC AGA AAT A-3'), *β-actin* (f: 5'-CCG TGC CAA TTT ATG AAG GGT AT-3', r: 5'-GAA GAT GAA GAG GCC GCA GTT T-3'), and *gapdh* (f: 5'-GAT GGG CAT GCT ATT TCG GTT TAT-3', r: 5'-CTT TGC TCG GTT GTT TTT GGT ATG-3'). Gene expression was normalized across the samples using *β-actin* and *gapdh* levels.

## Whole mount RNA in situ hybridization (WISH) and immunostaining protocol

For RNA expression analysis, worms were stained using previously established WISH protocols (*King and Newmark, 2013*) with additional modifications for increased reagent penetrance and washing efficiency. These modifications permitted effective in situ analysis of worm or worm fragments ranging from 1 to 8 mm in size. All washes were performed in 50 ml Falcon tubes or 2 ml tubes in accordance with worm numbers in place of 24 well plates. All incubations and washes were performed for 10 min with nutation at room temperature unless otherwise specified.

Mucus was removed with 7.5% NAC in PBS for 8 min. Worms were fixed in 4% PFA 0.3% Triton X for 30 min and washed twice with 1X PBS 0.3% Triton-X (PBSTx). Worms were then washed with 10% SDS for increased permeabilization. Reduction was performed at 37°C for 10 or 20 min with a solution of 1X PBS, 1% NP-40, 0.5% SDS, 50 mM DTT. Worms were washed twice with PBSTx 0.3%, and then dehydrated with a 50% MeOH: 50% PBSTx 0.3% solution, and washed and stored in 100% MeOH O/N at −20°C.

In preparation for sample bleaching, worms were rehydrated with a 50% MeOH: 50% PBStX 0.3% wash, washed twice with PBSTx 0.3%, and washed once in 1X SSC. For bleaching, worms were incubated in a 1.2% $H_2O_2$, 5% Formamide, 0.5X SCC solution for 4 hr over a light box. Worms were washed once with 1X SSC and twice with 1X PBS 0.3% Tween-20 (PBSTw 0.3%). Worms were permeabilized for 20 min with a 1X PBS 0.1% SDS solution containing 2–4 µg/ml Proteinase K (in accordance with worm size). Worms were post-fixed in 4% PFA 0.3% Tw for 10 min and washed twice with PBSTw 0.3%. Worms were prepared for hybridization with a 10 min 50% solution wash and subsequent 2 hr 56°C incubation with a prehybridization solution containing 50% Formamide, 1X Denhardts solution, 100 ug/ml Heparin, 1% Tween-20, 50 mM DTT, and 1 mg/mL Sigma Torula Yeast RNA in 5X SSC. DIG-labeled and/or DNP-labeled probes were denatured 5 min at 70°C, and incubated O/N at 56°C in a hybe solution identical to prehybe solution but with Deionized Formamide in place of Formamide, Calbiochem Yeast RNA 0.25 mg/ml in place of Sigma Torula Yeast RNA, and an addition of 5% Dextran Sulfate.

The next day non-specific probe binding was washed out at 56°C. The samples were washed twice for 30 min in a Wash Hybe solution containing 50% Formamide, 0.5% Tween-20, and 1% Denhardts in 5X SSC. Worms were then washed twice in 50% Wash Hybe:50% 2X SSC 0.1% Tw for 30 min, thrice with 2XSSC 0.1% Tw for 20 min, and thrice with 0.2X SSC 0.1% Tw for 20 min. The samples were returned to room temperature and washed twice with 1X MAB 0.1–03% Tw (MABTw) pH 7.5. The samples were blocked for 2 hr in a solution of 1X MABTw containing 10% Horse Serum and 0.5% Roche Western Blocking Reagent and stained with anti-DIG and/or anti-DNP Fab fragment conjugated to alkaline phosphatase (colorimetric development) or peroxidase (fluorescent development) at a 1:1000 dilution O/N at 4°C. For the final antibody incubation prior to development and slide mounting, this solution was supplemented with 1:5000 DAPI. Non-specific binding was removed with washing 6 times with MABTw for 2–3 hr and then processed for colorimetric or fluorescent development.

For colorimetric development, worms were incubated for 15 min in solutions of 0.1 M Tris pH 9.5, 0.1 M NaCl, 0.05 M MgCl₂, 0.1% Tween containing 0%, 50%, and then 80% PVA. The final solution of 80% PVA was supplemented with 1:188 BCIP or 1:94 NBT and worms monitored for color development. The reaction was stopped by rinsing 1–2 times in PBS, fixed in 4% PFA 0.3% Tx for 45 min, and washed 3 times with PBSTx 0.3%. The samples were then washed with 100% EtOH for 20 min and then with 50% EtOH:50% PBS for 5 min. The samples were the rehydrated with 1X PBS washes and mounted in 80% glycerol.

For fluorescent development, worms were pre-incubated for 15 min in a development solution of 0.1 M Boric Acid, 2 M NaCl, pH 8.5 supplemented at 1:2000 with tyramide. Worms were incubated another 45 min in this solution after the addition of 0.006% H2O2. Samples were washed twice with PBSTw 0.3% and peroxidase activity was terminated with a 1 hr incubation in 200 mM NaN3 in PBSTw 0.3%. The samples were washed twice in PBSTw 0.3% and either mounted in a 4 M Urea, 0.1% Triton-X, 2.5% DABCO, 20% glycerol solution. If additional immunostaining was performed, worms were washed 6 times in PBSTw 0.3% for 2–3 hr, 4 times with MABTw 0.1–0.3%, and then incubated O/N with the specified primary antibody.

## Unidirectional flow system culture

Planaria were cultured in paper cups (24oz ECO products containers or 9oz Dixie cups) lined with a pocket of 150 μm nylon mesh (Saatitech, Italy). Cups were perforated with holes at the desired planaria water level for outflow to complete the flow vessel. These flow vessels were placed in an enclosed container on a perforated tray suspended above a collection basin and centralized waste drain. Water was pumped from a 55 gallon tank into a series of 44 spigots directly above the flow vessels and adjusted to a constant drip rate of 1 drop per 5 s for 9oz cups and 1 drop per 2 s for 24oz cups. The system actively pumped water for 1.5 hr every 4 hr achieving total dispensation of approximately 55 gallons of planarian water every 3.5 days. Flow was turned off during RNAi feedings. Flow vessels were washed after RNAi feeding and worms were washed and transferred to fresh flow vessels the day before RNAi feeding or the start of experiments.

## Gene cloning and RNAi feeding experiments

Candidate genes analyzed in this study were cloned from a CIW4 cDNA library into a pPR-T4P vector (J. Rink) as described elsewhere (*Adler et al., 2014*). These served as a template for in vitro dsRNA synthesis for RNAi feedings (*Rouhana et al., 2013*). RNAi food was prepared by mixing 1 volume of dsRNA at 50–200 ng/μl (depending on the experiment) with 1.5 volumes of beef liver paste. The amount of food administered was typically 1 to 2 μl of food per worm depending on animal size. Worms were allowed to feed for 6 to 10 hr with 2 rounds of agitation by gentle pipetting to facilitate additional consumption. Worms were fed every 3 days for a total of 3 RNAi feedings. Experiments were conducted at 4 days or more after the last RNAi feeding.

## Data analysis of pathological scores in RNAi screen

Following enumeration of individual worm pathological scores, data for each day was reduced to an average pathological score. Days for each RNAi condition were aligned in ascending order along the y-axis of a column. The average score of each column was calculated and used to sort the effects of RNAi conditions in ascending order along the x-axis. For the frequency of individual stages, the percentage of worms exhibiting the corresponding score each day was plotted in place of the average score. Heat maps were generated using TM4 MeV. Significance was calculated using 2-WAY ANOVA, $p < 0.05$.

## P-p38 western blot analysis

Western blot analyses were performed as previously described (*Hatton et al., 2011*). The P-p38 antibodies used in this study were acquired from Cell Signaling Technologies. Both polyclonal (#9211) and monoclonal (#4511) antibodies were used for the initial assessment of cross-reactivity with the planarian phospho-epitope. Subsequent validation and in situ staining analysis was conducted with the monoclonal antibody due to higher in situ staining signal.

## TUNEL and phospho H3 or phospho p38 staining

Worms were prepared for TUNEL or immunostaining identically to those prepared for WISH up to the point of sample dehydration and storage in MeOH. Worms were rehydrated and then bleached in a solution of 3% $H_2O_2$, 0.075% $NH_4O_4$ in PBSTx for 5 hr over light. Worms were then washed 3X with PBSTx, permeabilized with 4 μg/ml for 10 min, post-fixed in 4% PFA PBSTx for 10 min, and washed another 3 times in PBSTx. For H3P and P-p38 staining, the samples were blocked and then stained for 48 hr at 4°C with primary anti-phospho H3 at 1:500 or anti-phospho p38 at 1:800 and then with secondary anti-rabbit HRP at 1:1000. For TUNEL labeling, the samples were stored O/N in PBS at 4°C.

TUNEL was performed as previously described (*Tu et al., 2015*). In brief, the samples were incubated in equilibration buffer for 15 min and then the TdT reaction was performed for 4 hr at 37°C. Afterwards, the samples were washed 6 times with PBSTx for 20 min, blocked, and incubated O/N at 4°C with anti-DIG POD at 1:1000. The samples were processed for fluorimetric development identically to those in the WISH protocol and either mounted or prepared for additional immunostaining.

## Acknowledgements

We would like to thank the members of the Sánchez Alvarado laboratory for their support. We thank Eric Ross and Kirsten Gotting for help with bioinformatics analysis and feedback, Boris Rubinstein for input on statistical analysis and data visualization, Kate Hall for RNA-seq library assistance, Michael Peterson for 16 s rDNA sequencing assistance, and Sharon Beckham and Nancy Thomas for histology assistance. We also acknowledge all other members of the Histology, Microscopy, Molecular Biology, and Reptile and Aquatics facilities at the Stowers Institute for their technical support. CPA is a post-doctoral fellow and ASA is an investigator of the Howard Hughes Medical Institute. This work was funded in part by NIH R37GM057260 to ASA.

## Additional information

### Competing interests

ASA: Reviewing editor, *eLife*. The other authors declare that no competing interests exist.

### Funding

| Funder | Grant reference number | Author |
| --- | --- | --- |
| Howard Hughes Medical Institute | | Alejandro Sánchez Alvarado |
| National Institute of General Medical Sciences | R37GM057260 | Alejandro Sánchez Alvarado |
| Stowers Institute for Medical Research | | Alejandro Sánchez Alvarado |

The funders had no role in study design, data collection and interpretation, or the decision to submit the work for publication.

### Author contributions

CPA, Conception and design, Acquisition of data, Analysis and interpretation of data, Drafting or revising the article; MSM, Was instrumental in helping design and build the new planarian culture systems reported in the manuscript, Conception and design, Contributed unpublished essential data or reagents; AH-A, SAM, SL, KNP, Acquisition of data, Analysis and interpretation of data; CWS, Conception and design, Analysis and interpretation of data; LG, Provided wild worm samples collected in Sardinia for analysis, Additionally aided in the experimental design and execution of assays for the appropriate sexual worms ; ASA, Conception and design, Analysis and interpretation of data, Drafting or revising the article

### Author ORCIDs

Alejandro Sánchez Alvarado, http://orcid.org/0000-0002-1966-6959

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
