## [Decision Letter]

Thank you for submitting your article "Pathogenic shifts in endogenous microbiota impede tissue regeneration via distinct activation of TAK1/MKK/p38" for consideration by *eLife*. Your article has been favorably evaluated by Richard Losick (Senior editor) and three reviewers, one of whom is a member of our Board of Reviewing Editors. The reviewers have opted to remain anonymous.

The reviewers have discussed the reviews with one another and the Reviewing Editor has drafted this decision to help you prepare a revised submission.

Summary:

In this manuscript, Arnold et al. demonstrate that planarians serve as a tractable model to study the relationship between the microbiome, immune system, and tissue regeneration due to their prolific regenerative abilities and the high degree of conservation of multiple immune-signaling genes. Arnold and colleagues take advantage of a novel low-septic culturing method they have developed in which planarians are first reared in a circulating water culture system and then transferred to traditional static cultures. This transition results in declining planarian health, triggered by bacterial pathogens, and manifests in easy-to-score tissue lesions, degeneration of tissue, and eventual lysis. The authors perform bacterial 16s rDNA deep sequencing under different culturing conditions to elucidate the composition and dynamics of the planarian microbiome. Interestingly, transition in culture conditions or tissue amputation elicits a reduction of Bacteroidetes and expansion of Proteobacteria (e.g., *Pseudomonas*) that correlates with susceptibility of worms to develop lesions. The strength of the paper lies in the methodology that Arnold et al. have devised for harnessing the shift in microbiome composition (caused by transitioning worms from circulating, low-septic culture methods to static culture) to identify host transcriptional changes by RNA Sequencing. The authors then use a candidate-based approach as proof of principle to examine conserved immune regulators and identify the TAK1/MKK/p38 signaling module as an important mediator of pathological progression.

Essential revisions:

1) Figure 1 – By 16s rDNA deep sequencing, the authors find that the microbiomes of wild-type and lab-grown planarians share many genera. The authors state that a significant proportion of bacterial genera is distinct in the wild-type samples, which could be attributed to either region-specific bacterial genera or the loss of certain genera during lab culture. To determine if the differences in microbiota are due to loss of certain genera during lab culture it would be more appropriate to compare the wild-type sexual planarians from Sardinia to the sexual lab strain instead of the asexual CIW4 strain. Are the wild-type sexuals more divergent in their microbiome than the two lab strains?

2) Figure 3 – Clustering analysis identified various transcriptional patterns, including one in which genes (e.g., *pgrp-1* and *pgrp-4*) were significantly upregulated when planarians transitioned to static culture. The authors should validate that the *pgrp* genes are induced by *Pseudomonas* infection by quantitative real-time PCR. WISH is not quantitative and the qualitative increase in *pgrp-4* expression in Figure 3 is not convincing. Is *pgrp-1* expression also upregulated? Based on how this manuscript is organized, many readers may well expect to see results from unbiased screening of some genes in this important data set. In fact, the IMD candidate lists could have been obtained without any of these transcriptomic data, so some readers may wonder why these data are included here if they will not be subject to more in-depth analysis. Are any of the other 32 IMD pathway homologs from the candidate RNAi screen present in this (or other) gene cluster(s)?

3) By RNAi, the authors convincingly demonstrate that certain genes are activators/inhibitors of the conserved IMD pathway by reducing/enhancing pathological progression in response to infection. However, the epistasis analysis performed by combinatorial RNAi experiments is confounding and certain aspects of the pathway are quite different from what has already been worked out in *Drosophila* and mammalian models. For example, the authors place planarian *cyld-1* epistatically downstream of Tak1, whereas in *Drosophila* and mouse models it acts upstream (CYLD is a deubiquitinating enzyme that physically interacts with Tak1 and regulates its ubiquitination). There is also equivalent evidence for placing Mkk4 upstream of PP6 rather than in parallel to the other genes, just as the other double RNAi outcomes were interpreted as indicative of a linear pathway. An explanation for such discrepancies could be that epistasis analysis using anything other than null alleles can be fraught with difficulties, especially when genes play roles in other developmental processes or if genetic redundancy is involved. Given the pitfalls of performing epistasis analysis by reduced-penetrance RNAi, the authors should either provide additional evidence supporting their divergent pathway or remove these data and the problematic pathway interpretations from the manuscript (Figure 6).

4) The identified activators/inhibitors could either influence infection clearance through an immune process or the host's ability to repair the damage caused by infection. Does inhibition of critical activators/inhibitors affect the levels of bacteria within the host during infection (for example, at early times in the progression of the pathology phenotypes)?

5) Are neoblasts depleted during normal infection-induced death or does the pathology mainly involve preventing neoblasts from repairing tissue quickly enough for recovery? This experiment would provide information about the mechanism of tissue pathology from infection.

6) Figure 8 – The authors should perform double FISH to show that components of the TAK1/MKK/p38 signaling module are expressed in the same cells in the gut and mesenchyme.

7) Figure 8 – p38 signaling dynamics data should be quantified and statistically analyzed. For example, it is not clear from the figures shown that the "increase in punctate p38^+^ cells" in response to infection (3dpi) is a significant change. Also, the authors should verify whether the "large nuclear dense mounds" they observe in the gut lumen consist of "P-p38^+^ cells" and not cellular debris. Should the cellular localization of P-p38^+^ signal be nuclear? In Figure 8 (5mpa), two of the three puncta (top- and bottom-most) do no co-localize with either nuclear DAPI or cytoplasmic mat1 staining. What does this P-p38^+^ signal represent? The authors should perform double immunofluorescence with a gut antibody and anti-P-p38 to analyze the P-p38^+^ cells at high resolution. It may be helpful for the readers if arrows are placed in each panel to show the location of P-p38^+^ signal. What does the intense *mat1*^+^ signal represent in the gut lumen post-amputation (5mpa)? Is this cellular debris or non-specific staining?

---

## [Author Response]

*Essential revisions:*

1) Figure 1 – By 16s rDNA deep sequencing, the authors find that the microbiomes of wild-type and lab-grown planarians share many genera. The authors state that a significant proportion of bacterial genera is distinct in the wild-type samples, which could be attributed to either region-specific bacterial genera or the loss of certain genera during lab culture. To determine if the differences in microbiota are due to loss of certain genera during lab culture it would be more appropriate to compare the wild-type sexual planarians from Sardinia to the sexual lab strain instead of the asexual CIW4 strain. Are the wild-type sexuals more divergent in their microbiome than the two lab strains?

We agree with the reviewers that our hypotheses for the source of the distinct genera of the wild sexual planarians did not take into account possible differences between the bacterial composition of sexual and asexual strains. To address this, we have sampled prominent culturable bacterial colonies from lab reared sexual worms. As no sexual strains are currently housed in recirculatory culture, we have taken worms from a fill and drain system which houses planaria strains in individual vessels and cycles planarian water. We sampled the culturable bacteria from worms 3 days after transition to static culture and found that they contained bacteria both similar to and distinct from those of asexuals (Figure 1—figure supplement 1). Therefore we have changed this section to the following:

*“*With respect to wild worms, this level of overlap was ~46%, suggesting that a significant proportion of bacterial genera is distinct in our wild sexual samples. […] Importantly, our analyses demonstrate that the vast majority of genera from CIW4 lab strain planaria are common to those of wild type sexual worms."

2) Figure 3 – Clustering analysis identified various transcriptional patterns, including one in which genes (e.g., pgrp-1 and pgrp-4) were significantly upregulated when planarians transitioned to static culture. The authors should validate that the pgrp genes are induced by Pseudomonas infection by quantitative real-time PCR. WISH is not quantitative and the qualitative increase in pgrp-4 expression in Figure 3 is not convincing. Is pgrp-1 expression also upregulated? Based on how this manuscript is organized, many readers may well expect to see results from unbiased screening of some genes in this important data set. In fact, the IMD candidate lists could have been obtained without any of these transcriptomic data, so some readers may wonder why these data are included here if they will not be subject to more in-depth analysis. Are any of the other 32 IMD pathway homologs from the candidate RNAi screen present in this (or other) gene cluster(s)?

We agree with the reviewers that qPCR is the most appropriate method for quantitative of gene changes in this case. We have therefore performed qPCR for *pgrp-1* and *pgrp-4*, confirming significant up-regulation of *pgrp-4* following *Pseudomonas* infection (Figure 3). We have also further clarified our rationale for our choice of the IMD pathway and homologous pathway components for our screen. In our RNAseq data the cluster with the clearest enrichment in genes that correlated with increase in bacteria (cluster 1) also contained upstream components of this pathway. This, in combination with published data, lead us to a characterization of this pathway in this process. From our RNAseq we did not observe significant transcriptional changes in the other members of this pathway at the sampled time points, but the reviewers attest that we have identified many of these genes are functionally involved in and/or post-transcriptionally modified (p38-1 phosphorylation) in response to bacterial infection. Therefore, the section has been changed to the following:

“Of particular interest was cluster 1, consisting of genes upregulated gradually after exit from the recirculation system in the absence of antibiotic (Figure 3). […] This data as well as the established role of this pathway in immunity and apoptosis lead us to take a candidate based investigation of the role of inflammatory signaling pathways in altered tissue homeostasis and regeneration during infection.”

*3) By RNAi, the authors convincingly demonstrate that certain genes are activators/inhibitors of the conserved IMD pathway by reducing/enhancing pathological progression in response to infection. However, the epistasis analysis performed by combinatorial RNAi experiments is confounding and certain aspects of the pathway are quite different from what has already been worked out in Drosophila and mammalian models. For example, the authors place planarian cyld-1 epistatically downstream of Tak1, whereas in Drosophila and mouse models it acts upstream (CYLD is a deubiquitinating enzyme that physically interacts with Tak1 and regulates its ubiquitination). There is also equivalent evidence for placing Mkk4 upstream of PP6 rather than in parallel to the other genes, just as the other double RNAi outcomes were interpreted as indicative of a linear pathway. An explanation for such discrepancies could be that epistasis analysis using anything other than null alleles can be fraught with difficulties, especially when genes play roles in other developmental processes or if genetic redundancy is involved. Given the pitfalls of performing epistasis analysis by reduced-penetrance RNAi, the authors should either provide additional evidence supporting their divergent pathway or remove these data and the problematic pathway interpretations from the manuscript* (*Figure 6*).

We agree with the reviewers that the evidence used to demonstrate epistasis in other more genetically tractable organisms and cell culture systems is more robust than our current methodologies. Therefore, it is difficult to resolve to what extent the differences relative to other species that we find in our analysis are attributable to distinct organization of this pathway in planaria versus the limitations of RNAi to effectively resolve these relationships. With respect to *cyld-1*, planaria in fact have four different *cyld* homologs. Therefore, while *cyld-1* was placed downstream of *tak1* and *jnk*, the position of *cyld-2, cyld-3*, or *cyld-4* could be more analogous to their hierarchy in Homo sapiens. Furthermore, in *Drosophila* CYLD functions in distinct manner relative to Homo sapiens. Rather than deubiquitinating TAK1 to regulate signaling, dCYLD deubiquitinates dTRAF2 to prevent proteolysis (Xue, Igaki, et.al. 2007). Given these differences between the roles of CYLD in Homo sapiens and *Drosophila*, and the additional CYLD homologues in planaria, we cannot conclude that the placement of *cyld-1* relative to the other components is inherently flawed. Additionally, we did acknowledge in the manuscript text that mkk4 could be placed either upstream or lateral to the other pathway components but realize this could be further clarified and emphasized in Figure 6.

To address these concerns we have altered this section and figure to more accurately reflect the appropriate conclusions we can draw from this analysis. Rather than designate this as an epistatic demonstration of the planarian TAK1 pathway, we have changed it to a combinatorial RNAi experiment to uncover the Phenotype Hierarchy of RNAi effects. Additionally, we have reorganized Figure 6 to clarify to what extent this Phenotype Hierarchy is similar to or distinct from that of Homo sapiens.

The text now reads:

“The results of our analyses allowed us to order these components with respect to their roles in mediating anterior tissue in response to infection: *mkk4, pp6, tak1, jnk, ppm1b, cyld-1, jun D, mkk6-1*, and *p38-1*. […] Thus, it remains to be determined to what extent the phenotype hierarchy uncovered here reflects the true *S. mediterranea* TAK1 signaling cascade.”

4) The identified activators/inhibitors could either influence infection clearance through an immune process or the host's ability to repair the damage caused by infection. Does inhibition of critical activators/inhibitors affect the levels of bacteria within the host during infection (for example, at early times in the progression of the pathology phenotypes)?

To address the potential roles of activators/inhibitors in the immune response we infected worms with *Pseudomonas* and tracked both bacterial levels and pathological progression over time for both control, and candidate activator/inhibitor RNAi. This analysis revealed that (1) the planarian immune response is coordinated with but not the result of induced tissue degeneration and apoptosis, and (2) jun D (part of the downstream branch of our Phenotype Hierarchy) has a role in immune defense. We have added a new section and figure (Figure 10) to the manuscript:

“Downstream TAK1 signaling components have a role in the planarian immune response

We evaluated the potential role of TAK1 signaling pathway homologs in the planarian immune response. […] Thus, *Pseudomonas* infectioninduces a *jun D* and *p38-1* dependenttissue degeneration response in conjunction with a *jun D* dependent, *p38-1* independent immune response.”

5) Are neoblasts depleted during normal infection-induced death or does the pathology mainly involve preventing neoblasts from repairing tissue quickly enough for recovery? This experiment would provide information about the mechanism of tissue pathology from infection.

To address this we analyzed neoblast levels via WISH of the marker piwi during a time course of *Pseudomonas* infection to look for overt effects on neoblast numbers over time. We found that while there was no obvious decrease in neoblast numbers we did observe some evidence that neoblast localization was altered during infection (less near the interior gut and more towards the periphery). We are following up on these results and have made the appropriate additions and modifications to the manuscript to reflect these findings.

Results section:

“Given that *Pseudomonas* infection progressively compromised tissue homeostasis (Figure 2), we next determined whether neoblasts and regeneration potential were similarly affected. We analyzed neoblast levels and distribution using WISH analysis. While infected worms exhibited abundant numbers of *piwi*^+^ neoblasts both prior to and following tissue degeneration, the frequency of neoblasts within the interior of the animal relative to the periphery appeared to progressively decrease during infection (Figure 2—figure supplement 1).”

Discussion section:

“Planaria may utilize TAK1 innate immune signaling for a coordinately mounting of antibacterial responses while actively clearing infected tissue via apoptosis prior to the initiation of regeneration. […] Under this context attenuation of infection-induced TAK1/MKK/p38 signaling rescued tissue degeneration and restored regenerative potential.”

6) Figure 8 – The authors should perform double FISH to show that components of the TAK1/MKK/p38 signaling module are expressed in the same cells in the gut and mesenchyme.

We have performed double fluorescent WISH experiments and the results provide further evidence of the co-expression of TAK/MKK/p38 signaling components in the gut and mesenchyme (Figure 8—figure supplement 1).

“WISH analysis of members of the TAK1/MKK/p38 signaling pathway revealed that these components were expressed in largely overlapping patterns of tissues along the planarian body axis. […] Weaker staining was also visible in the mesenchymal space between the gut and the epidermis with a slight anterior enrichment (Figure 8, Figure 8—figure supplement C).”

*7) Figure 8 – p38 signaling dynamics data should be quantified and statistically analyzed. For example, it is not clear from the figures shown that the "increase in punctate p38^+^ cells" in response to infection (3dpi) is a significant change. Also, the authors should verify whether the "large nuclear dense mounds" they observe in the gut lumen consist of "P-p38^+^ cells" and not cellular debris. Should the cellular localization of P-p38^+^ signal be nuclear? In Figure 8 (5mpa), two of the three puncta (top- and bottom-most) do no co-localize with either nuclear DAPI or cytoplasmic mat1 staining. What does this P-p38^+^ signal represent? The authors should perform double immunofluorescence with a gut antibody and anti-P-p38 to analyze the P-p38^+^ cells at high resolution. It may be helpful for the readers if arrows are placed in each panel to show the location of P-p38^+^ signal. What does the intense mat1^+^ signal represent in the gut lumen post-amputation (5mpa)? Is this cellular debris or non-specific staining?*

We have quantified and statistically analyzed P-p38 signaling (Figure 8). We have also provided a higher resolution Z-series of the anterior gut demonstrating the co-localization of P-p38 signal with nuclei in the gut at 5 minutes post amputation (Figure 8—figure supplement 3). The evidence of intact nuclei in close proximity to the observed signal is inconsistent with the reviewer’s hypothesis that this represents cellular debris. Additionally, previous studies have demonstrated that cytoplasmic p38 can localize to the nucleus in response to certain stimuli such as DNA damage (Wood, Thornton, et. Al 2009). As planarian gut specific antibodies are not commercially available and we do not currently have one that is compatible with our P-p38 staining protocol, we have instead performed P-p38 staining in combination with FISH of 4 independent gut RNA probes (*mat1, porc, hnf4*, and *nkx-2.2*) to address the reviewer’s concern (Figure 8). Arrows have been placed as a visual aid.

“We compared the kinetics of P-p38 activation in response to injury versus bacterial infection. […] Quantitative image analysis confirmed that while infection resulted in an increase in P-p38 fraction volume over time, amputation induced a sharp increase in both P-p38 fraction volume and average intensity followed by a gradual decrease (Figure 8).”